# Hepatitis C Virus NS5A Activates Mitophagy Through Cargo Receptor and Phagophore Formation

**DOI:** 10.3390/pathogens13121139

**Published:** 2024-12-23

**Authors:** Yuan-Chao Hsiao, Chih-Wei Chang, Chau-Ting Yeh, Po-Yuan Ke

**Affiliations:** 1Department of Biochemistry & Molecular Biology, Graduate Institute of Biomedical Sciences, College of Medicine, Chang Gung University, Taoyuan 33302, Taiwan; c11kkh@gmail.com (Y.-C.H.); will0521@gmail.com (C.-W.C.); 2Liver Research Center, Chang Gung Memorial Hospital, Taoyuan 33305, Taiwan; chauting@cgmh.org.tw

**Keywords:** HCV, HCV NS5A, mitophagy, cargo receptor, phagophore

## Abstract

Chronic HCV infection is a risk factor for end-stage liver disease, leading to a major burden on public health. Mitophagy is a specific form of selective autophagy that eliminates mitochondria to maintain mitochondrial integrity. HCV NS5A is a multifunctional protein that regulates the HCV life cycle and may induce host mitophagy. However, the molecular mechanism by which HCV NS5A activates mitophagy remains largely unknown. Here, for the first time, we delineate the dynamic process of HCV NS5A-activated PINK1/Parkin-dependent mitophagy. By performing live-cell imaging and CLEM analyses of HCV NS5A-expressing cells, we demonstrate the degradation of mitochondria within autophagic vacuoles, a process that is dependent on Parkin and ubiquitin translocation onto mitochondria and PINK1 stabilization. In addition, the cargo receptors of mitophagy, NDP52 and OPTN, are recruited to the mitochondria and required for HCV NS5A-induced mitophagy. Moreover, ATG5 and DFCP1, which function in autophagosome closure and phagophore formation, are translocated near mitochondria for HCV NS5A-induced mitophagy. Furthermore, autophagy-initiating proteins, including ATG14 and ULK1, are recruited near the mitochondria for HCV NS5A-triggered mitophagy. Together, these findings demonstrate that HCV NS5A may induce PINK1/Parkin-dependent mitophagy through the recognition of mitochondria by cargo receptors and the nascent formation of phagophores close to mitochondria.

## 1. Introduction

Regular targeting of cytosolic components for lysosomal degradation through autophagy ensures nutrient recycling and promotes intracellular organelle regeneration [1,2]. The loss of controlled autophagy may contribute to the development and pathogenesis of several human diseases [1,2]. The completion of autophagy involves the arrangement of intracellular membranes and the biogenesis of vacuoles to target intracellular materials for lysosomal degradation [3,4,5]. For the initiation of autophagy, the suppression of the mammalian target of rapamycin complex 1-induced endoplasmic reticulum (ER) translocation of the unc-51-like kinase (ULK) complex (containing ULK1/2, autophagy-related gene 101 [ATG101], ATG13, and FIP200/RB1CC1) activates and recruits the class III phosphatidylinositol-3-OH kinase (PI3KC3) complex I (containing Vps34, Vps15, ATG14, and Beclin 1) for the synthesis of phosphatidylinositol-3-phosphate (PtdIns(3)P) on the ER-associated microdomain. Newly-generated PtdIns(3)P leads to the translocation of WD-repeat domain PtdIns(3)P-interacting (WIPI) family proteins and double-FYVE-containing protein 1 (DFCP1) for phagophore emergence [3,4,5]. Two ubiquitin-like conjugation systems, consisting of the formation of the ATG12-ATG5-ATG16 trimeric complex and phosphatidylethanolamine (PE) conjugation to the C-terminus of the ATG8/microtubule-associated protein light chain 3 (LC3) protein (referred to as ATG8/LC3 lipidation), promote the expansion and closure of phagophores into mature double-membranous autophagosomes [3,4,5]. Subsequently, autophagosomes fuse with lysosomes, leading to the formation of autolysosomes, which contain acidic hydrolases that degrade sequestered materials [3,4,5]. Finally, the termination of autophagy-reactivated mammalian target of rapamycin (mTOR) drives the lysosome reformation and the recycling of autophagosomal components, thereby promoting the regeneration of lysosomes and autophagosomes [6,7].

Not only for a bulk and non-selective degradation, autophagy also selectively eliminates intracellular cargoes, including organelles and proteins, in a process referred to as “selective autophagy” [8,9]. Selective autophagy is a type of intracellular organelle quality control, so-called “organellophagy”, that maintains the integrity of organelles and promotes organelle regeneration [10,11]. Mitophagy is a specific form of organellophagy that removes damaged mitochondria, thus promoting mitochondrial turnover [12,13]. The mitochondrial depolarization–stabilization of the mitochondrial kinase, PTEN-induced putative kinase 1 (PINK1) on the mitochondrial outer membrane (MOM) recruits the Parkinson protein 2, E3 ubiquitin protein ligase (Parkin) to mitochondria, and induces the phosphorylation of ubiquitin and Parkin at serine (Ser) 65 [12,13]. Parkin subsequently triggers the protein ubiquitination of MOM proteins [12,13], and this process promotes the recognition of ubiquitinated mitochondria by selective autophagy cargo receptors, such as calcium-binding and coiled-coil domain-containing protein 2 (Calcoco2/NDP52) and optineurin (OPTN) [14]. By interacting with ATG8/LC3 via the LC3-interacting region (LIR), these mitophagy receptors ultimately target deformed mitochondria to the autophagic process for turnover [12,13]. In recent years, several studies have shown that the phosphorylation of mitophagy receptors by tank-binding kinase 1 (TBK1), and the formation of phagophores driven by the autophagy initiation machinery, are coordinated with the initiation of PINK1/Parkin-dependent mitophagy [15,16,17,18]. Phagophore-resident phosphatidylinositol 4,5-bisphosphate (PtdIns(4,5)P_2_) coordinates with the disassembly of mitoaggregates and the initiation of autophagy to activate PINK1/Parkin-dependent mitophagy [15]. The ATG13, a component of the ULK1 complex, can be recruited to ubiquitinated mitochondria after mitophagy receptor recognition and act as a coordinator for phagophore emergence on ER strands wrapped around mitochondria, whereas FIP200 functions independently upstream of the recruitment of mitophagy receptors to ubiquitinated mitochondria [16]. Additionally, the TBK1-mediated phosphorylation of mitophagy receptors, the PI3KC3 complex I, and the ULK1 complex may promote the initiation of PINK1/Parkin-dependent mitophagy [17,18].

Hepatitis C virus (HCV) is the most prevalent enveloped RNA virus, with more than 3% of the human population being infected; HCV infection leads to chronic liver diseases, and ultimately to hepatocellular carcinoma (HCC) [19,20]. Direct-acting antiviral drugs (DAAs) are currently used to treat HCV infection, with a curing rate of 90% [21,22]. Notably, the emergence of drug-resistant variants in patients after DAA treatment impedes the complete eradication of HCV infection [23,24]. Like other RNA viruses, HCV may activate host cellular autophagy to promote viral growth and escape the innate immune defense system [25,26]. In addition, HCV infection can trigger selective autophagy to promote the catabolism of intracellular organelles, including mitochondria and lipid droplets (LDs), thereby promoting the establishment of persistent infection and preventing the excess accumulation of lipids [25,26]. The HCV genome comprises a single-strand and positive-sense RNA with a length of approximately 9.6 kilobases, which contains untranslated regions at the 5′- and 3′-termini and a single open reading frame (ORF) [27,28]. The ORF of HCV viral RNA encodes a single polypeptide of 3300 amino acids (a.a.), which is processed by host cellular and viral proteases [27,28], thus generating three structural proteins (core, envelope glycoprotein 1 [E1], and E2) for the constitution of the infectious virion and seven nonstructural (NS) proteins (p7, NS2, NS3, NS4A, NS4B, NS5A, and NS5B) for the replication of viral RNA and the assembly of infectious particles [27,28].

HCV NS5A is a hyperphosphorylated protein that functions in regulating viral RNA replication and virion assembly through interactions with viral RNA, NS4B, NS5B, and host cellular proviral factors [29,30]. Additionally, the phosphorylation of multiple Ser and threonine (Thr) residues of HCV NS5A regulates the replication of the viral genome and the production of infectious virions [29,30]. In addition, HCV NS5A may regulate mitochondrial biogenesis and the turnover of mitochondria via autophagy [31,32]. The ectopic expression of HCV NS5A may induce ER membrane tethering around mitochondria and promote mitochondrial fission via the kinase activity of phosphatidylinositol 4-kinase IIIα (PI4KA) to protect cells from cell apoptosis [31]. However, the overexpression of HCV NS5A can induce autophagy and enhance autophagic flux, as demonstrated by increased PE-conjugated LC3 and the accumulation of autophagosomes and autolysosomes [32]. Moreover, HCV NS5A overexpression induces mitochondrial depolarization and the localization of endogenous Parkin to mitochondria [32], suggesting that HCV NS5A can activate mitophagy, presumably in a Parkin-dependent manner. However, the molecular mechanism by which HCV NS5A activates mitophagy remains largely unknown. Additionally, the dynamics of HCV NS5A-induced mitophagy, such as the sequestration of mitochondria by autophagic vacuoles and the mitochondrial translocation of Parkin, are not fully understood. Most importantly, whether the recruitment of mitophagy receptors and autophagy-initiating machinery for phagophore formation function in HCV NS5A-induced mitophagy is unclear.

In this study, we elucidate the molecular process of HCV NS5A-induced PINK1/Parkin-dependent mitophagy, in which NDP52 and OPTN are necessarily involved. In addition, our findings provide the first line of evidence showing that autophagy initiation molecules are recruited to the vicinity of mitochondria for phagophore formation in the HCV NS5A-activated PINK1/Parkin-dependent mitophagy.

## 2. Materials and Methods

### 2.1. Cell Culture, Chemicals, and Antibodies

The human hepatoma cell line Huh7 was kindly provided by Charles M Rice (Rockefeller University, New York, NY, USA) and Francis Chisari (Scripps Research Institute, San Diego, CA, USA). Human embryonic kidney 293 cells, in which the SV40 large T antigen was transformed (referred to as HEK293T, CRL-3216), were obtained from the American Type Cell Collection (ATCC, Manassas, VA, USA). Huh7/NDP52 knockout (KO), Huh7/OPTNKO, and Huh7/Parental cell lines were established with clustered regularly interspaced short palindromic repeats (CRISPR) genome editing in our previous study [33]. Huh7 and HEK293T cells were cultured in DMEM (Thermo Fisher Scientific, Waltham, MA, USA) supplemented with 10% (*v*/*v*) fetal bovine serum (FBS, Thermo Fisher Scientific), 1% (*v*/*v*) penicillin-streptomycin-glutamine (PSG, Thermo Fisher Scientific), and 1% (*v*/*v*) nonessential amino acids (NEAAs, Thermo Fisher Scientific) at 37 °C in a 5% CO_2_ atmosphere. Hoechst 33342 (H1399), used for IF, was purchased from Thermo Fisher Scientific. Sodium cacodylate buffer (CAB, 0.4 M) (11655), osmium tetroxide (OsO_4_), (19190), lead citrate (17800), uranyl acetate (22400), paraformaldehyde (PFA), EM grade (157–158, 4% [*v*/*v*] in water), glutaraldehyde (GA), EM grade (16220, 25% [*v*/*v*] in water), and a low-viscosity embedding medium Spurr’s resin kit (14300), used for the fixation, embedding, and staining of samples for electron microscopy, were purchased from Electron Microscopy Sciences (Hatfield, PA, USA). Ethanol (1.00983.2500) and potassium hexacyanoferrate(II) trihydrate (K_4_[Fe(CN)_6_]·3H_2_O, 455989), which were used for dehydration and fixation for electron microscopy, were obtained from Merck Millipore and Sigma–Aldrich (St. Louis, MO, USA) and Merck Millipore (Burlington, MA, USA), respectively. Rabbit anti-phospho-ubiquitin (Ser65) (ab309155), mouse anti-mitochondrially encoded cytochrome C oxidase II (MTCO2) (ab110258), and Rabbit anti-PINK1 (ab23707) antibodies were purchased from Abcam (Cambridge, UK). Goat anti-rabbit Alexa Fluor 647 (A-21244) and horseradish peroxidase (HRP)-conjugated secondary antibodies (A0545 and A9044, respectively) were obtained from Thermo Fisher Scientific and Sigma–Aldrich, respectively. Rabbit anti-mitofusin-2 (MFN2) (9482) and mouse anti-Parkin (4211) antibodies were purchased from Cell Signaling Technology (Burlington, MA, USA). Mouse anti-translocase of outer mitochondrial membrane 20 (TOMM20) (sc-17764) and mouse anti-cytochrome c oxidase subunit 4 (COX4) (sc-376731) antibodies were obtained from Santa Cruz (Dallas, TX, USA). Mouse anti-β-actin (A5441) antibody and mitochondrial division inhibitor 1 (Mdivi-1) (M0199) were purchased from Sigma (Aizu, Japan). The small interference RNA (siRNA) duplexes against human Parkin were obtained from Invitrogen Stealth RNAi collection.

### 2.2. Plasmid Construction

pTRIP-RFP-LC3, pTRIP-Mito-GFP, pTRIP-Mito-RFP, pTRIP-Mito-miRFP670, pLenti-EF-YFP-Parkin, pLenti-EF-RFP-Parkin, pLenti-EF-GFP-NDP52, pLenti-EF-GFP-OPTN, pTRIP-mitochondrial quality control (Mito-QC; RFP-GFP-FIS1 [101–152]), and pTRIP-mitochondria-targeted Keima-Red (MT-Keima) expression plasmids were constructed as described previously [33]. To construct pmTagBFP2-N1-HCV NS5A and pmiRFP670-N1-HCV NS5A, a polymerase chain reaction (PCR)-amplified HCV NS5A gene fragment from the HCV JFH1 infectious clone (genotype 2a, kindly provided by Takaji Wakita [National Institute of Infectious Disease, Tokyo, Japan] [34]) was subcloned and inserted into pmTagBFP2-N1 (#54566, Addgene) and pmiRFP670-N1 (#79987, Addgene). The gene fragments containing HCV NS5A-mTagBFP2 and HCV NS5A-miRFP670 were subsequently subcloned and inserted into the pTRIP-GFP lentiviral plasmid (kindly provided by Charles M Rice [35]), generating pTRIP-HCV NS5A-mTagBFP2 and pTRIP-HCV NS5A-miRFP670, respectively. PCR-amplified HCV core and NS5B were subcloned and inserted into pmiRFP670-C1, a modified pEGFP-C1 plasmid in which GFP was replaced by miRFP670. The gene fragments containing miRFP670-HCV core and miRFP670-HCV NS5B were subcloned and inserted into the pTRIP-GFP lentiviral plasmid, generating pTRIP-miRFP670-HCV core and pTRIP-miRFP670-HCV NS5B, respectively. PCR-amplified ubiquitin (Ub) from pRK5-HA-Ubiquitin (#17608, Addgene) and the PINK1 gene fragments from pEYFP-N1-PINK1 (#101874, Addgene) were subcloned and inserted into pEGFP-C1 (Takara, Osaka, Japan) and pmiRFP670-N1 to generate pEGFP-C1-Ub and pmiRFP670-N1-PINK1. The gene fragment containing EGFP-Ub was excised from pEGFP-C1-Ub, subcloned, and inserted into pTRIP-GFP, generating pTRIP-GFP-Ub. The PCR-amplified PINK1-miRFP670 gene fragment from pmiRFP670-N1-PINK1 was subcloned and inserted into pLenti-EF-Blast [33], generating pLenti-EF-Blast PINK1-miRFP670. All the sequences of the gene fragments used for plasmid construction in this study were confirmed via Sanger sequencing. Moloney murine leukemia virus (MMLV)-based retrovirus expression plasmids, including pMXs-IP-EGFP-mAtg5 (#38196), pMXs-puro GFP-DFCP1 (#38269), pMXs-IP GFP-Atg14 (#38264), and pMXs-IP-EGFP-ULK1 (#38193) were obtained from Addgene (Watertown, MA, USA).

### 2.3. Generation of Lentiviruses and Retroviruses and Virus-Mediated Gene Delivery

The generation of lentiviruses for this study was performed as described previously [36]. To generate pTRIP-derived lentiviruses, the pTRIP-cDNA transfer plasmid containing the gene of interest, pCMVdeltaR8.91 (a packaging plasmid containing the gag, pol, and rev genes of human immunodeficiency virus [HIV], obtained from the RNAi core facility, Academia Sinica, Taibei, Taiwan), and pMD.G (a vesicular stomatitis virus [VSV]-G envelope-expressing plasmid obtained from the RNAi core facility) were cotransfected at a ratio of 1:1:1 into HEK293T cells. For the production of pLenti-EF-related lentiviruses, the pLenti-EF-cDNA transfer plasmid containing the gene of interest, psPAX2 (a packaging plasmid harboring the HIV gag, pol, and rev genes, #12260, Addgene) and pMD2.G (a VSVG-expressing plasmid, #12259, Addgene) were cotransfected as described above. For the generation of pMX-derived retroviruses, the pMX expression plasmid harboring cDNA was cotransfected with pHCMV-AmphoEnv (an amphotropic MMLV Env-expressing plasmid, #15799, Addgene) at a ratio of 1:1 into 293T cells. After seventy-two hours, culture supernatants containing infectious lentiviruses and retroviruses were harvested, filtered through a 0.45-μm filter, and used for transduction. For gene delivery through lentivirus and retrovirus transduction, Huh7 cells were seeded at a density of 2 × 10^5^ cells/well in a 12-well plate and then transduced with lentiviruses and retroviruses supplemented with 8 μg/mL polybrene via the spin inoculation method (1100× *g*, 4 h at 25 °C). After twenty-four hours, the lentiviruses and retroviruses were removed, and fresh DMEM containing 10% FBS, 1% NEAAs, and 1% PSG was added.

### 2.4. Confocal Microscopy and Live-Cell Imaging

For confocal microscopy, cells seeded on a coverslip were washed twice with phosphate-buffered saline (PBS, pH 7.4) and fixed with 4% PFA in PBS for 30 min at room temperature (RT). After extensive washing with PBS, the cells were imaged using a Plan-Apochromat 63x/1.4 Oil DIC M27 instrument coupled with a laser scanning confocal microscope (LSM780, Zeiss, Jena, Germany). The images were processed and analyzed using Zen 3.0 blue edition software (Zeiss). The number of colocalizations between autophagic vacuoles and mitochondria, the mitochondrial translocation of Parkin, the translocation of ubiquitin and cargo receptors, and the colocalization of PINK1 on autophagic vacuoles containing mitochondria and Parkin-translocated mitochondria, were analyzed with Zen 3.0 blue software and quantified manually. The quantification of the number of mitolysosomes (Mito-QC reporter) and the measurement of mitophagic flux (MT-Keima reporter) were performed via confocal microscopy and software as described previously [33,37,38]. The graphs of each quantified dataset shown in this study were generated with GraphPad Prism 9.0 software, and statistical significance was analyzed via a two-tailed *t*-test with a confidence interval of 95%. For time-lapse live-cell imaging, the cells were plated on 35-mm high ibiTreat μ-Dishes (ibidi, Gräfelfing, Germany) at a density of 2 × 10^5^ cells. Twelve hours after seeding, the cells were incubated in a complete medium (DMEM containing 10% FBS, 1% NEAAs, and 1% PSG) in an enclosed incubator (37 °C with 5% CO_2_) equipped with an LSM780 laser scanning confocal microscope. Time-lapse images of live cells were continuously captured for three hours at 1-min intervals by a Plan-Apochromat 63x/1.4 oil DIC M27 objective. The live-cell images and the serial frames of each movie were generated and assessed using Zen 3.0 blue edition software.

### 2.5. Correlative Light and Electron Microscopy

Correlative light and electron microscopy (CLEM) was performed according to a previously-described procedure [33,39,40,41]. Briefly, cells were seeded into poly-D-lysine-coated, 35-mm gridded glass-bottom cell culture dishes (MatTek P35G-1.5-14-CGRD) pretreated with poly-D-lysine (A3890401, Thermo Fisher Scientific) at a density of 5 × 10^4^, and incubated with DMEM supplemented with 10% FBS, 1% NEAAs, and 1% PSG for twenty-four hours. Then, the cells were washed twice with PBS (pH 7.4) and once with 0.1 M CAB buffer, pH 7.2. After extensive washing with 0.1 M CAB buffer (pH 7.2), the cells were fixed with 0.1 M CAB buffer (pH 7.2) containing 0.5% GA and 4% PFA for thirty minutes, at 37 °C. The cells were then washed twice with 0.1 M CAB buffer (pH 7.2), stored in one milliliter of 0.1 M CAB buffer (pH 7.2), and imaged using a Plan-Apochromat 63x/1.4 Oil DIC M27 instrument equipped with an LSM780 laser scanning confocal microscope. Images of twenty stacks (200 nm/stack) of different Z-positions of the cells of interest were captured and processed into three-dimensional (3-D) reconstitution images using Zen 3.0 blue edition software. After analysis via confocal microscopy, the cells were fixed with 0.1 M CAB buffer (pH 7.2) containing 1% OsO_4_ and 15 mg/mL K_4_[Fe(CN)_6_] for one hour at RT. After two washes with 0.1 M CAB buffer (pH 7.2) and three washes with double distilled water (ddH_2_O), the cells were incubated with 0.4% uranyl acetate for one hour at 4 °C and washed extensively with ddH_2_O. After dehydration in a graded series of ethanol (50%, 70%, 90%, and 100%), the cells were embedded in Spurr’s resin, and the embedded samples were trimmed and sectioned with an EM UC7 ultramicrotome. Approximately twenty serial sections (70 nm/section) were collected and stained with 2% uranyl acetate for ten minutes at RT and then incubated with 0.4% lead citrate solution for six minutes at RT. Electron micrographs were captured using a transmission electron microscope (JEM 1230, Joel, Tokyo, Japan) at a magnification of 10,000 at 100 kV, assembled into one montage, and used for the relocation of the position of cells of interest on the Z-stack images of the confocal micrographs. The montage of electron micrographs and the defined section of the Z-stack images were aligned and analyzed using Adobe Photoshop CS6 software.

### 2.6. SDS-PAGE and Western Blotting

To harvest protein extracts from cells, the cells were washed twice with PBS and lysed with RIPA buffer (50 mM Tris, pH 7.4; 150 mM NaCl; 1% NP-40; 0.5% DOC; and 0.1% SDS) containing protease inhibitors (Roche, Kaiseraugst, Switzerland) and phosphatase inhibitors (Sigma). The cell lysate was clarified by centrifugation at 12,000× *g* at 4 °C for 10 min to harvest the soluble protein extract. After the concentration of protein extract was determined via a Bradford protein assay (Bio-Rad, Hercules, CA, USA), equal amounts of protein were separated via SDS–PAGE and then transferred onto a 0.45-μm PVDF membrane (Millipore). The transfer membrane was subsequently blocked with TBST buffer (20 mM Tris, pH 7.4; 150 mM NaCl; and 0.1% Tween-20) supplemented with 3% nonfat milk. Thirty minutes later, the transfer membrane was incubated with primary antibodies and cognate secondary antibodies conjugated to HRP. Finally, an enhanced chemiluminescence kit (Millipore) was used to detect the protein signals on the membrane according to the manufacturer’s instruction manual.

## 3. Results

### 3.1. HCV NS5A Induces the Sequestration of Mitochondria Within Autophagic Vacuoles

Previous studies have shown that HCV NS5A can trigger mitochondrial fission, mitochondrial depolarization, and endogenous Parkin translocation to mitochondria [31,32], suggesting that HCV NS5A may cause mitophagy. However, the molecular process by which HCV NS5A regulates hepatic mitophagy is poorly understood. Here, we combined confocal microscopy and time-lapse live-cell imaging to assess the process of mitophagy induced by HCV NS5A in Huh7 human hepatoma cells, which have been shown to be permissive to HCV infection. As shown in Figure 1A (top panel) and Figure 1B, the ectopic expression of HCV NS5A-mTagBFP2 by lentiviral gene delivery significantly increased the number of RFP-LC3 (a conventional fluorescence reporter used for analyzing the formation of autophagic vacuoles) [42]-labeled autophagic vacuoles, which contained a population of mitochondria expressing Mito-GFP, a fluorescence reporter containing GFP fused with the mitochondrial targeting sequence (MTS) human cytochrome c subunit VIII oxidase (COX8A). In contrast, there were no detectable RFP-LC3-labeled autophagic vacuoles and no significant colocalized signals between RFP-LC3 and Mito-GFP in the cells without HCV NS5A-mTagBFP2 expression (Figure 1A; bottom panel and Figure 1B).

In addition, time-lapse live-cell imaging captured the dynamic engulfment of Mito-GFP-marked mitochondria by RFP-LC3-labeled autophagic vacuoles in HCV NS5A-expressing cells (Figure 1C; indicated by white arrowheads and Appendix A). Moreover, we employed CLEM to ultrastructurally evaluate the sequestration of mitochondria by HCV NS5A-induced autophagic vacuoles. The confocal micrographs of different Z-stacks of HCV NS5A-expressing cells, as shown in Figure 1D, were assembled, deconvoluted, and reconstituted into a 3-D structure (Figure 1E), which revealed that numerous RFP-LC3-labeled autophagic vacuoles encompassed Mito-GFP-containing mitochondria. Furthermore, the aligned confocal and CLEM micrographs revealed that the area in which RFP-LC3 and Mito-GFP were colocalized contained a population of autophagic vacuoles that sequestered deformed mitochondria (Figure 1F; indicated by white arrowheads in the magnified electron micrographs), suggesting that HCV NS5A expression induced the engulfment and elimination of mitochondria in mitophagosomes. In particular, the CLEM image in Figure 1G (indicated by white arrows) shows phagophores wrapped around the mitochondria. These results suggest that HCV NS5A may activate mitophagy to promote the degradation of mitochondria.

### 3.2. HCV NS5A Triggers the Translocation of Parkin to Mitochondria

Given that the recruitment of Parkin to mitochondria is critical for promoting the ubiquitination of proteins on the MOM to initiate PINK1/Parkin-dependent mitophagy [43], we next assessed whether the mitochondrial translocation of Parkin is necessary for HCV NS5A-induced mitophagy. As shown in Figure 2A,B, a more significant amount of RFP-Parkin colocalized with Mito-GFP-labeled mitochondria in HCV NS5A-miRFP670-expressing cells (top panel in Figure 2A) compared with that observed in cells lacking HCV NS5A-miRFP670 (Figure 2A; bottom panel). Moreover, time-lapse live-cell imaging of HCV NS5A-miRFP670-expressing RFP-Parkin/Mito-GFP cells revealed the substantial translocation of RFP-Parkin to mitochondria labeled with Mito-GFP (Figure 2C; indicated by white arrowheads and Appendix A). Similarly, a CLEM image of YFP-Parkin/Mito-RFP cells harboring HCV NS5A-miRFP670i revealed that YFP-Parkin was recruited to Mito-RFP-expressing mitochondria (Figure 2D,E). Furthermore, CLEM ultrastructural analysis of HCV NS5A-miRFP670i-expressing YFP-Parkin/Mito-RFP cells revealed that YFP-Parkin and Mito-RFP merged, where numerous autophagic vacuoles sequestered degradative mitochondria (Figure 2F; indicated by arrowheads in magnified electron microscopy images), directly indicating that Parkin translocation to mitochondria is involved in HCV NS5A-activated mitophagy. Notably, phagophores were observed in the region proximal to the mitochondria in the HCV NS5A-miRFP670i-expressing YFP-Parkin/Mito-RFP cells; the phagophore exhibited early-stage autophagosome formation and Parkin translocation to mitochondria (Appendix A; indicated by white arrows in the magnified electron micrograph in [D]). These results suggest that HCV NS5A can induce the mitochondrial translocation of Parkin to drive mitophagy.

### 3.3. HCV NS5A Activates Mitophagy to Degrade Mitochondria

Although HCV NS5A was previously shown to induce mitochondrial depolarization and the mitochondrial translocation of endogenous Parkin, it remains uncertain whether HCV NS5A can induce mitochondrial degradation through mitophagy. The mitophagy reporter Mito-QC contains a tandem fluorescence tag (RFP-GFP) fused with human mitochondrial fission 1 (FIS1) MTS (a.a. 101–152), which has been used for the semiquantitative measurement of mitophagic degradation [44,45]. The RFP and GFP signals of the Mito-QC reporter remained stable in mitochondria (RFP^+^/GFP^+^; yellow colored) in the neutral environment of the cytosol. When mitochondria are sequestered within autophagic vacuoles for degradation, the GFP signal of the Mito-QC reporter is quenched by low lysosomal pH, but the RFP signal of the Mito-QC reporter emits predominantly; these effects can be used for detecting mitolysosome formation (RFP^+^/GFP^−^; red colored). Using this reporter, we found that HCV NS5A-mTagBFP2 expression significantly increased the number of RFP^+^/GFP^−^-mitolysosomes in cells (Figure 3A; top panel and Figure 3B); in contrast, HCV NS5A-deficient cells presented abundant RFP^+^/GFP^+^ signals (Figure 3A; bottom panel). In addition, time-lapse live-cell imaging of HCV NS5A-mTagBFP2-expressing Mito-QC cells revealed the formation of RFP^+/^GFP^−^-mitolysosomes (Figure 3C; indicated by white arrowheads and Appendix A). Moreover, CLEM analysis of HCV NS5A-mTagBFP2-expressing Mito-QC cells revealed the presence of RFP^+/^GFP^−^-mitolysosomes, in which degradative mitochondria were engulfed (Figure 3D–F; indicated by white arrowheads in the magnified electron micrographs).

We further utilized MT-Keima, a mitophagy reporter harboring a Keima fluorescence tag fused with human FIS1 MTS, to assess HCV NS5A-triggered mitophagy degradation, as previously reported [33,38,46]. A low pH-dependent shift from excitation at a shorter wavelength (440 nm, in a neutral environment) to excitation at a longer wavelength (561 nm, in an acidic environment) for the MT-Keima reporter allows for the assessment of mitophagic flux [33,38,46]. We found that HCV NS5A-BFP2 expression triggered the dominant expression of the MT-Keima reporter at 561 nm (Figure 4A; top panel and Figure 4B). In contrast, no significant fluorescence signal of the 560 nm-excited MT-Keima reporter was detected in cells lacking HCV NS5A (the bottom panel in Figure 4A,B), indicating that HCV NS5A enhanced the mitophagic flux of MT-Keima in cells. Live-cell imaging further revealed a time-lapse increase in the fluorescence signal of MT-Keima excited at 561 nm in HCV NS5A-BFP2-expressing cells (indicated by arrowheads in Figure 4C and Appendix A), confirming again that HCV NS5A induced the degradation of mitochondria. In contrast to HCV NS5A, HCV core and HCV 5B expression did not induce detectable mitophagic cells (Appendix A). In addition, treatment with a mitophagy inhibitor, Mdivi-1 [47], and gene knockdown of endogenous Parkin significantly inhibited the HCV NS5A-enhanced mitophagic flux (Appendix A). These results indicate that HCV NS5A specifically activates Parkin-dependent mitophagy to promote mitochondrial turnover.

### 3.4. HCV NS5A Induces the Translocation of Ubiquitin and the Stabilization of PINK1 on Mitochondria

Since PINK1 stabilization on the MOM and subsequent recruitment of ubiquitin to mitochondria for the ubiquitination of MOM proteins are prerequisites for initiating PINK1/Parkin-dependent mitophagy, we next analyzed whether HCV NS5A activates mitophagy by inducing the mitochondrial translocation of ubiquitin and the stabilization of PINK1 on the MOM. The confocal micrographs and quantitative data in Figure 4D,E show that the ectopic expression of HCV NS5A induced an increase in the amount of GFP-Ub recruited to Mito-miRFP670-expressing mitochondria, which were sequestered by RFP-LC3-labeled autophagic vacuoles. In addition, time-lapse live-cell imaging revealed a dynamic process in which GFP-Ub translocated onto Mito-miRFP670-labeled mitochondria, followed by subsequent engulfment by RFP-LC3 puncta (indicated by arrowheads in Figure 4F and Appendix A). Moreover, there was a significant increase in the level of phosphorylated Ub (Ser65) on RFP-LC3-labeled autophagy-sequestered Mito-GFP-mitochondria in HCV NS5A-mTagBFP2-expressing cells compared with no detectable phosphorylation of Ub at Ser65 in cells lacking HCV NS5A (Figure 5A,B). Additionally, the colocalization of phospho-Ub (Ser65) with RFP-Parkin-translocated Mito-GFP-mitochondria was similarly increased by the expression of HCV NS5A-mTagBFP2, as shown in Figure 5C,D. These studies imply that the recruitment and phosphorylation of Ub are necessary for HCV NS5A-induced mitophagy.

Furthermore, we found that HCV NS5A expression triggered the stabilization of PINK1-miRFP670 on Mito-GFP-expressing mitochondria, which were sequestered by RFP-LC3-labeled autophagic vacuoles (Figure 5E; top panel and Figure 5F) and translocated by RFP-Parkin (Figure 5G; top panel and Figure 5H). In contrast, no apparent colocalization of PINK1-miRFP670 with RFP-LC3 (the bottom panel in Figure 5E) or RFP-Parkin (the bottom panel in Figure 5G) onto Mito-GFP was detected, suggesting that HCV NS5A expression triggered the stabilization of PINK1 on mitochondria, resulting in the initiation of mitophagy. This phenomenon was confirmed by the increased level of PINK1 protein in the mitochondrial fraction of HCV NS5A-expressing cells (Appendix A). In addition, HCV NS5A expression was shown to increase the protein ubiquitination of MFN2 in mitochondria, a hallmark of PINK1/Parkin-dependent mitophagy [48,49], and to downregulate the protein expression of mitochondrial proteins, including MTCO2, MFN2, COX4, and TOMM20 (Appendix A). These results suggest that HCV NS5A may induce PINK1/Parkin-dependent activity by stabilizing PINK1 and inducing mitochondrial ubiquitination.

### 3.5. HCV NS5A Activates PINK1/Parkin-Dependent Mitophagy Through NDP52 and OPTN

Several mitophagy receptors have been shown to recognize ubiquitinated mitochondria for the turnover of PINK/Parkin-mediated mitophagy [50,51]. Our previous study demonstrated that NDP52 and OPTN act as mitophagy receptors of hepatic mitophagy through the PINK1/Parkin pathway [33]. To date, the mitophagy receptor responsible for the HCV NS5A-induced degradation of mitochondria remains unidentified. Thus, we used confocal microscopy to analyze whether NDP52 and OPTN can be recruited to the mitophagic process in HCV NS5A-expressing cells. We found that GFP-NDP52 and GFP-OPTN significantly translocated onto RFP-LC3 puncta where the Mito-miRFP670-labeled mitochondria were sequestered in cells harboring HCV NS5A-mTagBFP2, whereas no overlapping signals of GFP-NDP52 and GFP-OPTN with RFP-LC3 puncta and Mito-miRFP670 were detected in cells lacking HCV NS5A (GFP-NDP52 shown in Figure 6A,B; GFP-OPTN shown in Figure 6C,D), suggesting that NDP52 and OPTN were specifically recruited to the HCV-NS5A-induced mitophagosomes.

Moreover, HCV NS5A-mTagBFP2 expression led to the significant recruitment of GFP-NDP52 and GFP-OPTN to Mito-miRFP670-expressing mitochondria, which contained a large amount of translocated RFP-Parkin (GFP-NDP52: top panel in Figure 6E,F; GFP-OPTN: top panel in Figure 6G,H). In contrast, there was no apparent colocalization of GFP-NDP52 and GFP-OPTN with RFP-Parkin or Mito-miRFP670 in cells without HCV NS5A expression (bottom panels of Figure 6E,G), suggesting the specific translocation of NDP52 and OPTN to Parkin-mediated mitophagy induced by HCV NS5A. Further live-cell imaging revealed that HCV NS5A induced the timely recruitment of GFP-NDP52 and GFP-OPTN to Mito-miRFP670-expressing mitochondria before being sequestered by RFP-LC3-labeled autophagic vacuoles (Appendix A). In addition, GFP-NDP52 and GFP-OPTN translocated to Mito-miRFP670-labeled mitochondria after the translocation of RFP-Parkin (Appendix A). Moreover, gene knockout of endogenous NDP52 (NDP52KO) and OPTN (OPTNKO) in cells led to a significant suppression in the HCV NS5A-induced formation of mitolysosomes (Appendix A) and the enhancement of mitophagic flux (Appendix A). These results indicate that NDP52 and OPTN serve as mitophagy receptors that recognize degradative mitochondria in HCV NS5A-activated PINK1/Parkin-dependent mitophagy.

### 3.6. Proximal Nascent Formation of Autophagosomes to Mitochondria for HCV NS5A-Induced PINK1/Parkin-Dependent Mitophagy

The interaction of LIRs within mitophagy receptors with ATG8/LC3 proteins on the membranes of pre-existing autophagosomes has been proposed to facilitate mitophagy; however, recent studies have demonstrated that molecules that function in early-stage autophagy, such as DFCP1, WIPI1, ATG5, and ATG16, can be recruited to damaged mitochondria before ATG8/LC3 conjugates to autophagosomal membranes during mitophagy [14,52,53]. To date, the molecular process of the nascent formation of autophagosomes in HCV- and HCV NS5A-induced mitophagy is not yet understood. Our CLEM data specifically showed the formation of phagophores near mitochondria in HCV NS5A-expressing cells (the electron micrographs are shown in Figure 1G and Appendix A). Since the ATG12-ATG5-ATG16L trimeric complex on phagophores promotes the expansion of phagophores to facilitate autophagosome formation for autophagy [5,54,55,56,57,58,59,60,61,62,63,64], we utilized confocal microscopy and live-cell imaging to monitor the biogenesis of nascent phagophores labeled with GFP-ATG5 during HCV NS5A-induced mitophagy. In the top panels of Figure 7A,B, we found that GFP-ATG5 mainly relocalized onto RFP-LC3 puncta containing Mito-miRFP670-labeled mitochondria. In contrast, no colocalization of GFP-ATG5 with RFP-LC3 and Mito-miRFP670 was detected in cells lacking HCV NS5A (Figure 7A; bottom panel). In addition, time-lapse live-cell image analysis revealed the translocation of GFP-ATG5 to Mito-miRFP670-expressing mitochondria before sequestration by RFP-LC3-labeled autophagic vacuoles (indicated by white arrowheads in Figure 7C and Appendix A), suggesting the proximal maturation of phagophores to autophagosomes in HCV NS5A-triggered mitophagy. Moreover, we further demonstrated that, compared with that in cells without HCV NS5A expression, GFP-ATG5 was recruited to RFP-Parkin-translocated Mito-miRFP670-labeled mitochondria in cells expressing HCV NS5A-mTagBFP2 (Figure 7D,E). The subsequent recruitment of GFP-ATG5 to Mito-miRFP670-expressing mitochondria after RFP-Parkin translocation in time-lapse live-cell imaging of HCV NS5A-mTagBFP2-expressing cells is shown in Figure 7F and Appendix A, again confirming the nascent formation of autophagosomes around degradative mitochondria in HCV NS5A-triggered mitophagy. These results support the notion that the formation of nascent autophagosomes near mitochondria is responsible for HCV-induced PINK1/Parkin-dependent mitophagy.

DFCP1, a PtdIns(3)P-binding protein, is recruited to the ER-associated subdomain for phagophore emergence and subsequent autophagosome assembly [65]. A recent study indicated that DFCP1 functions in the constriction of phagophores for selective autophagy, including mitophagy, through its ATPase activity [66]. On the basis of these studies, we assessed whether DFCP1 can be recruited to mitochondria for phagophore emergence to initiate HCV NS5A-activated mitophagy. The confocal micrographs in Figure 8A,B show the specific recruitment of DFCP1 to mitophagosomes containing RFP-LC3 and Mito-miRFP670 in HCV NS5A-mTagBFP2-expressing cells. Time-lapse live-cell imaging demonstrated that DFCP1 translocated to Mito-miRFP670-labeled mitochondria in a timely manner before RFP-LC3 puncta sequestered Mito-miRFP670-expressing mitochondria in HCV NS5A-induced mitophagy (indicated by white arrowheads in Figure 8C and Appendix A). In addition, the recruitment of GFP-DFCP1 to Mito-miRFP670-expressing mitochondria, in which RFP-Parkin translocated in cells harboring HCV NS5A-mTagBFP2, was demonstrated by confocal microscopy (Figure 8D,E). The live-cell images in Figure 8F and Appendix A confirmed that GFP-DFCP1 was dynamically recruited to RFP-Parkin-translocated Mito-miRFP670-labeled mitochondria in HCV NS5A-mTagBFP2-expressing cells. These results suggest that the translocation of DFCP1 to the region proximal to mitochondria may promote the emergence of phagophores to enclose mitophagosomes for HCV-induced PINK1/Parkin-dependent mitophagy.

### 3.7. The Recruitment of the Autophagy Initiation Machinery near Mitochondria for HCV NS5A-Induced PINK1/Parkin-Dependent Mitophagy

Recent studies have suggested that the autophagy initiation machinery, including PI3KC3 and ULK1, can be recruited to damaged mitochondria to induce the generation of phagophores, which further mature into autophagosomes in PINK1/Parkin-dependent mitophagy [16,53,67,68]. By assessing the subcellular localization of GFP-ATG14 and GFP-ULK1, we monitored the HCV NS5A-induced translocation of PI3KC3 and ULK1 during mitophagy. In the top panels of Figure 9A,B, we found that the ectopic expression of HCV NS5A-mTagBFP2 induced the recruitment of GFP-ATG14 to mitophagosomes, which contained merged signals of Mito-miRFP670-labeled mitochondria and RFP-LC3 puncta, in contrast to the homogenous distribution of GFP-ATG14 within the cytosol of cells lacking HCV NS5A (bottom panel). Time-lapse live-cell imaging revealed that GFP-ATG14 was recruited to Mito-miRFP670-expressing mitochondria beyond the sequestered RFP-LC3-labeled autophagic vacuoles (Figure 9C; indicated by white arrowheads and Appendix A). In addition, the colocalization of GFP-ATG14 with the RFP-Parkin-translocated Mito-miRFP670-labeled mitochondria was analogous (Figure 9D; top panel and Figure 9E). In contrast, there was no significant colocalization of GFP-ATG14 with RFP-Parkin or Mito-miRFP670 in cells not expressing HCV NS5A (Figure 9D; bottom panel and Figure 9E). The dynamic recruitment of GFP-ATG14 to Mito-miRFP670-expressing mitochondria after the mitochondrial translocation of RFP-Parkin in HCV NS5A-mTagBFP2-expressing cells was depicted by the time-lapse live-cell imaging in Figure 9F (as indicated by white arrowheads) and Appendix A. These results suggest that PI3KC3 was recruited to damaged mitochondria to initiate the formation of phagophores for HCV NS5A-induced PINK1/Parkin-dependent mitophagy.

In addition to GFP-ATG14, GFP-ULK1 was also recruited to mitophagosomes, in which RFP-LC3 puncta engulfed Mito-miRFP670-labeled mitochondria (Figure 10A,B) and RFP-Parkin-translocated Mito-miRFP670-expressing mitochondria (Figure 10D,E). Moreover, time-lapse live imaging studies revealed the timely recruitment of GFP-ULK1 to Mito-miRFP670-containing mitochondria before sequestration by RFP-LC3 puncta (Figure 10C; white arrowheads and Appendix A) and the subsequent recruitment of GFP-ULK1 after the mitochondrial translocation of RFP-Parkin to Mito-miRFP670-labeled mitochondria (Figure 10F; white arrowheads and Appendix A). These results imply that the recruitment of the ULK complex to damaged mitochondria is a prerequisite step to initiate de novo phagophore formation for PINK1/Parkin-dependent mitophagy in HCV NS5A-expressing cells.

## 4. Discussion

HCV NS5A positively regulates HCV growth at multiple layers, including enhancing the replication of viral RNA, promoting the assembly of infectious particles, and mediating the interaction of host proviral factors with HCV viral proteins [29,30]. Intriguingly, the exact enzyme activity of HCV NS5A remains unclear, although domain I is capable of zinc coordination [29,30]. Many studies have shown that HCV NS5A may alter mitochondrial biogenesis and induce autophagy to eliminate depolarized mitochondria [31,32]. However, the underlying mechanism is not yet understood, except for the mitochondrial localization of endogenous Parkin. Through time-lapse live-cell imaging and CLEM analysis, we reveal the molecular process of HCV NS5A-induced PINK1/Parkin-dependent mitophagy at the dynamic and ultrastructural levels. Our study demonstrates that HCV NS5A activates mitophagy to sequester mitochondria, which is where Parkin is translocated (Figure 1 and Figure 2). Also, HCV-induced PINK1/Parkin-dependent mitophagy promotes mitochondrial turnover (Figure 3 and Figure 4), and this process requires PINK1 stabilization, the recruitment of Ub to mitochondria, and the phosphorylation of Ub at Ser65 (Figure 5 and Appendix A). In addition, NDP52 and OPTN are recruited to mitochondria (Figure 6 and Appendix A) and act as the specific receptors of HCV-induced PINK1/Parkin-dependent mitophagy (Appendix A). Furthermore, the recruitment of autophagy-initiating machinery to mitochondria drives the nascent generation of phagophores for autophagosome biogenesis for HCV-activated PINK1/Parkin-dependent mitophagy (Figure 7, Figure 8, Figure 9 and Figure 10). Collectively, these results reveal that HCV NS5A induces ubiquitin-dependent mitophagy via the PINK1/Parkin pathway and that this process requires cargo receptor recognition and the recruitment of effectors for phagophore emergence.

How HCV NS5A triggers mitochondrial damage to activate PINK1/Parkin-dependent mitophagy remains enigmatic. It has been shown that the overexpression of HCV NS5A in HEK293T cells leads to mitochondrial fragmentation in a similar fashion to that in HCV Jc1 (genotype 2)-infected cells; this process requires the kinase activity of PI4KA [31,69], an HCV-NS5A-interacting protein that can promote the viral life cycle [70,71]. Notably, this HCV NS5A-induced fragmentation of mitochondria does not require dynamin-related protein 1 (Drp1) [31], a small GTPase that regulates mitochondrial fission [72]. The results of recent studies suggest that phosphatidylinositol 4-phosphate (PtdIns(4)P), a product of PI4KA, positively regulates the fission of mitochondria [73,74] and autophagosome maturation [75,76], suggesting that HCV NS5A may induce mitochondrial fission and autophagosome biogenesis through its binding to PI4KA, thereby activating mitophagy. Therefore, whether PI4KA is required for HCV NS5A-induced PINK1/Parkin and the underlying mechanism requires further investigation. Additionally, determining which domain of NS5A can induce mitophagy is of interest. The *n*-terminal region of HCV NS5A contains an amphipathic α-helix (AH) capable of interacting with intracellular membranes, such as the ER, which is necessary to promote HCV replication [29,30]. Whether the membrane anchoring ability of HCV NS5A AH applies to mitochondria, thus inducing mitochondrial deformation to initiate mitophagy, should be investigated. In addition, it would be interesting to examine whether the ability of zinc coordination within the highly-structured domain I of HCV NS5A is involved in the induction of mitophagy. In addition, whether the PI4KA-interacting motif of HCV NS5A domain I (a.a. 187–214 [77]) can bridge the association of PtdIns(4)P with mitochondria to trigger mitochondrial fission, thereby initiating mitophagy, should be investigated.

To date, the physiological significance of HCV NS5A-induced PINK1/Parkin-dependent mitophagy in the HCV life cycle and host cellular response is largely unknown. HCV-triggered mitochondrial fission can protect cells from apoptosis [31]. Additionally, HCV NS5A-induced mitophagy can be triggered by intracellular reactive oxygen species (ROS), and this process seems to serve as an antioxidant response [32]. Additionally, HCV JFH1 infection has been reported to activate mitophagy through the mitochondrial translocation of Parkin and Drp1 phosphorylation (Ser616)-induced mitochondrial fission [69,78], and HCV-induced mitophagy may attenuate cell apoptosis and promote viral persistence [69]. Accordingly, investigating whether HCV NS5A induces PINK1/Parkin-dependent mitophagy to promote mitochondrial turnover and protect cells from ROS-triggered cell apoptosis is worthwhile. In recent years, a functional role of mitophagy in the regulation of innate immunity has emerged [79,80]. For example, influenza A virus PB1-F2 and nucleoprotein proteins were shown to induce mitophagy to promote mitochondrial antiviral signaling protein (MAVS) degradation [81,82], thus repressing the type I interferon (IFN) antiviral response. It remains uncertain whether HCV NS5A can activate mitophagy to decrease the mitochondrial mass and/or promote the degradation of mitochondria-associated innate immune effectors, such as MAVS, to attenuate IFN antiviral immunity. Further studies are worthwhile to evaluate the physiological significance of HCV NS5A-activated mitophagy in the regulation of anti-HCV IFN response. In addition, whether HCV NS5A-activated PINK/Parkin-dependent mitophagy participates in the pathogenesis of HCV-related liver diseases, such as liver steatosis and HCC, is not yet known. Notably, mitophagy activation may promote the emergence of cancer stem cells and reprogram intracellular metabolism to drive the development of HCC [83,84]. In the future, the functional role(s) of HCV NS5A-induced mitophagy in the development of liver steatosis and HCC needs to be further investigated.

It remains unclear whether HCV NS5A induces PINK1/Parkin-dependent mitophagy, which can similarly reflect that of HCV-infected cells harboring a complete viral life cycle. The HCV core protein has been shown to mediate the relocalization of HCV NS proteins, including NS5A, on LD surfaces, promoting the assembly of infectious particles [85]. Additionally, the interaction between the HCV core and NS5A on the surface of LDs links the association of host proviral factors with HCV NS5A, including Rab18, diacylglycerol acyltransferase-1 (DGAT1), and perilipin 3 [86,87,88]. In addition, the HCV core has also been reported to interfere with HCV NS5A-induced mitophagy, although the detailed mechanism remains unclear [32]. Accordingly, it is questionable whether the expression of HCV NS5A alone remains near LDs in cells lacking the HCV core protein. Further studies are needed to compare and delineate the molecular process of mitophagy in HCV-infected cells and HCV subgenomic replicon cells capable of replicating viral RNA and lacking structural genes, including the core gene. HCV NS5A is a hyperphosphorylated protein in which several residues can be phosphorylated by host cellular kinases [29,30]. Therefore, whether HCV NS5A phosphorylation regulates PINK1/Parkin-dependent mitophagy is worthy of investigation. In addition to mitophagy, HCV NS5A was recently shown to induce other forms of autophagy, including CMA and endosomal microautophagy, to degrade hepatocyte nuclear factor 1 alpha and DGAT1, respectively [89,90,91], implying that HCV NS5A may simultaneously regulate multiple autophagy pathways. The molecular mechanism underlying the spatiotemporal regulation of HCV NS5A to induce these different types of autophagy requires further exploration.

In summary, our study provides a comprehensive model for the molecular process of HCV NS5A-induced PINK1/Parkin-dependent mitophagy, in which NDP52 and OPTN serve as mitophagy receptors and autophagy initiator machineries are recruited to mitochondria. The results provide insight into the regulatory mechanism of virus-induced mitophagy, enabling a better understanding of the possible role(s) of mitophagy in the pathogenesis of HCV-associated liver diseases.

## Figures and Tables

**Figure 1 pathogens-13-01139-f001:**
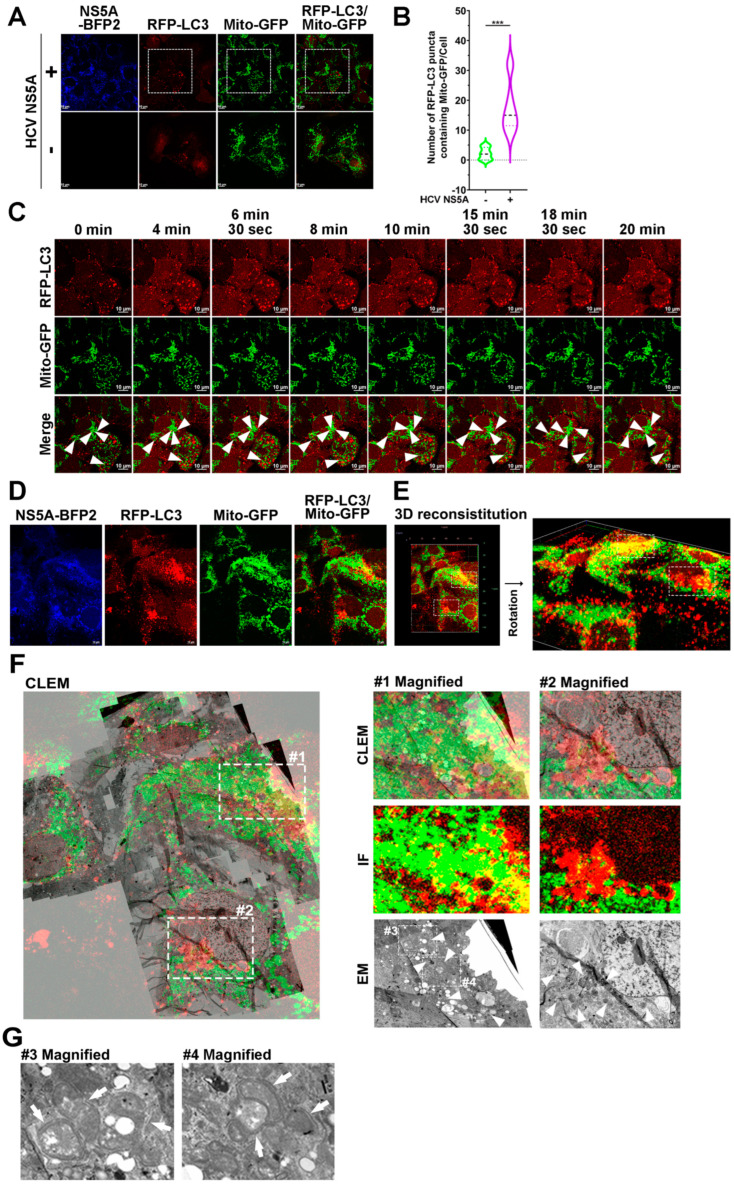
HCV NS5A induces the engulfment of mitochondria within autophagic vacuoles: (**A**) Huh7 cells were transduced with lentiviruses expressing RFP-LC3 and Mito-GFP, as described in the “Materials and Methods” section, to generate Huh7/RFP-LC3/Mito-GFP cells. Then, Huh7/RFP-LC3/Mito-GFP cells were transduced with (+) or without (−) pTRIP-HCV NS5A-mTagBFP2 lentiviruses for forty-eight hours and analyzed via confocal microscopy. (**B**) The degree of colocalization between RFP-LC3-labeled autophagic vacuoles and Mito-GFP-expressing mitochondria was quantified. The data are presented as means ± SEMs (*n* = 10, *** *p* < 0.001). (**C**) The selected live imaging frames show the magnified area in the white dashed box of the top panel in (**A**). The white arrowheads indicate the sequestration of Mito-GFP-labeled mitochondria by RFP-LC3 puncta. (**D**–**G**) CLEM analysis of mitochondrial sequestration by autophagic vacuoles in HCV NS5A-expressing cells. (**D**) Huh7/RFP-LC3/Mito-GFP cells were transduced with lentiviruses expressing HCV NS5A-mTagBFP2 for forty-eight hours and then processed for confocal microscopy. (**E**) The assembled Z-stacks of the confocal micrographs in (**D**) were reconstituted into a 3-D image. The white dashed boxes indicate the engulfment of Mito-GFP-expressing mitochondria within RFP-LC3 puncta. (**F**) The aligned image of the confocal micrograph (IF) and electron micrograph (EM) from the CLEM analysis of cells in (**D**) is shown. The white dashed boxes in the left panel are enlarged and shown in the magnified images in the right panel. The white arrowheads indicate the sequestration of mitochondria within autophagic vacuoles. (**G**) The enlarged images show the magnified white dashed boxes in the EM of (**F**). The white arrows indicate the phagophores wrapped around deformed mitochondria.

**Figure 2 pathogens-13-01139-f002:**
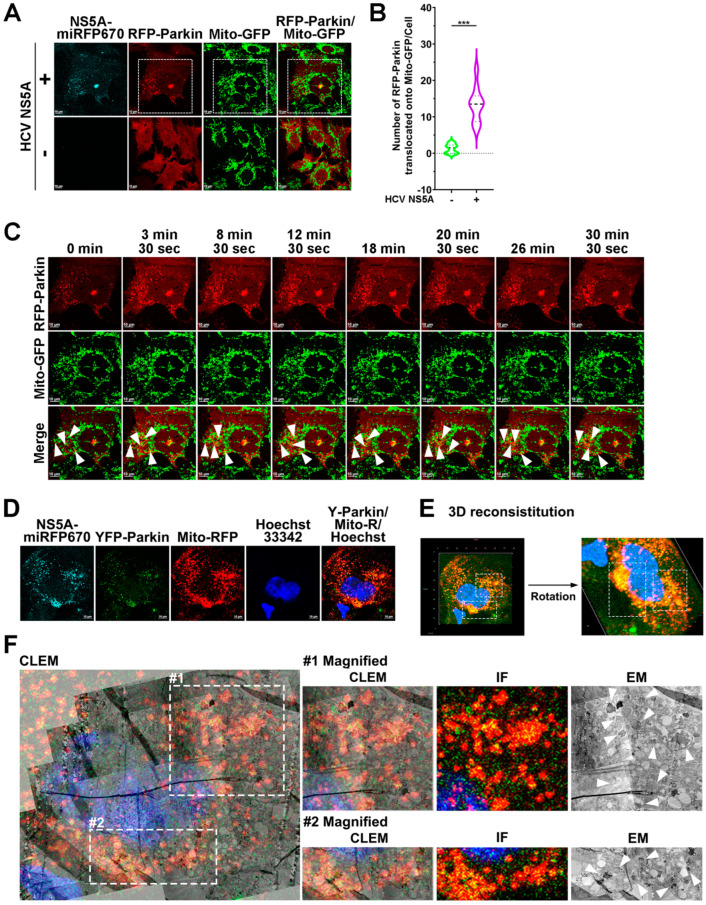
HCV NS5A induces the translocation of Parkin to mitochondria: (**A**) Huh7 cells were transduced with lentiviruses harboring RFP-Parkin and Mito-GFP, according to the procedure described in the “Materials and Methods” section, to establish Huh7/RFP-Parkin/Mito-GFP cells. Huh7/RFP-LC3/Mito-GFP cells were transduced with (+) or without (−) pTRIP-HCV NS5A-miRFP670 lentiviruses for forty-eight hours and then analyzed via confocal microscopy. (**B**) The degree of colocalization between RFP-LC3-Parkin and Mito-GFP-labeled mitochondria was quantified. The data are presented as means ± SEMs (*n* = 10, *** *p* < 0.001). (**C**) The selected live imaging frames show the magnified area in the white dashed box of the top panel in (**A**). The white arrowheads indicate the translocation of RFP-Parkin to Mito-GFP-expressing mitochondria. (**D**–**F**) CLEM analysis of the mitochondrial translocation of RFP-Parkin in HCV NS5A-expressing cells. (**D**) Huh7/RFP-Parkin/Mito-GFP cells were transduced with lentiviruses expressing HCV NS5A-miRFP670. Forty-eight hours later, the cells were analyzed via confocal microscopy. (**E**) The Z-stacks of the confocal micrograph shown in (**D**) were assembled and deconvoluted into a 3-D image. The white dashed boxes indicate the Mito-GFP-expressing mitochondria with RFP-Parkin translocation. (**F**) The aligned IF and CLEM image of the cells from (**D**) is presented. The white dashed boxes in the left panel are enlarged and shown in the magnified images in the right panel. The white arrowheads indicate the degradative mitochondria in which Parkin translocates.

**Figure 3 pathogens-13-01139-f003:**
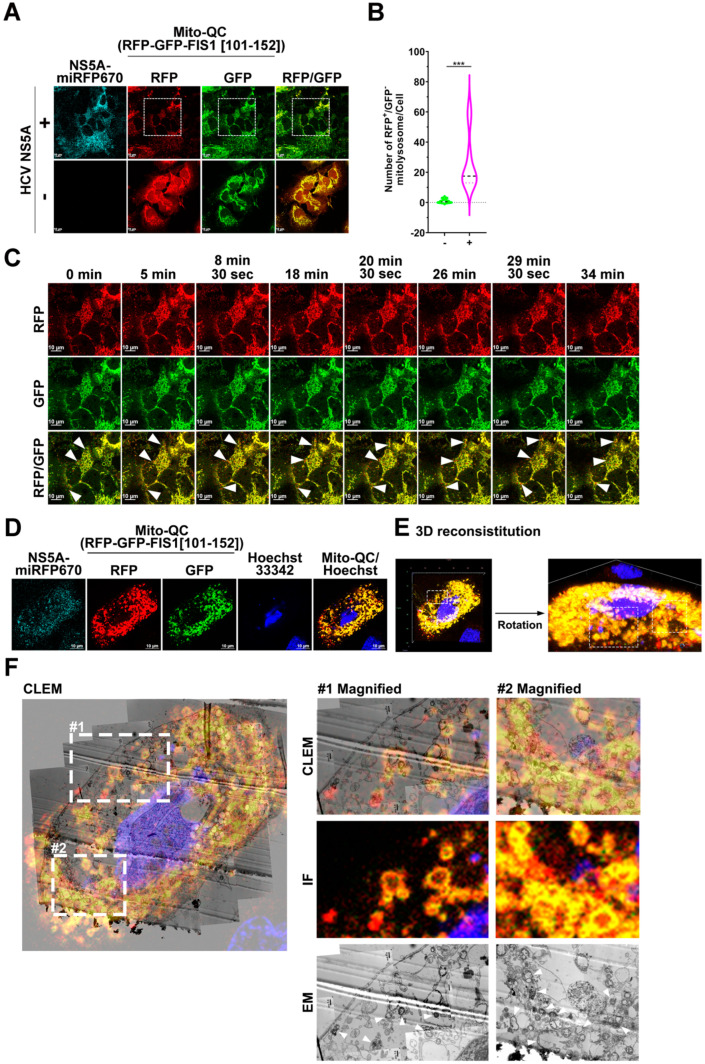
HCV NS5A induces mitolysosome formation: (**A**) Huh7 cells were transduced with pTRIP-Mito-QC lentiviruses, as described in the “Materials and Methods” section, generating Huh7/Mito-QC cells. Huh7/Mito-QC cells were transduced with (+) or without (−) pTRIP-HCV NS5A-miRFP670 lentiviruses. After forty-eight hours, the cells were analyzed via confocal microscopy. (**B**) The number of RFP^+^/GFP^−^ mitolysosomes was quantified with Image J, as described previously [33]. The data are presented as means ± SEMs (*n* = 10, *** *p* < 0.001). (**C**) The selected live imaging frames show the magnified area in the white dashed box of the top panel in (**A**). The white arrowheads indicate the formation of RFP^+^/GFP^−^ mitolysosomes. (**D**–**F**) CLEM analysis of mitolysosome formation in HCV NS5A-expressing cells. (**D**) Huh7/Mito-QC cells were transduced with lentiviruses expressing HCV NS5A-miRFP670 for forty-eight hours and then processed for confocal microscopy. (**E**) The Z-stacks of the confocal micrograph shown in (**D**) were assembled and deconvoluted into a 3-D image. The white dashed boxes indicate the loci of RFP^+^/GFP^−^ mitolysosomes. (**F**) The aligned IF and CLEM image of the cells from (**D**) is shown. The white dashed boxes in the left panel are enlarged and shown in the magnified images in the right panel. The white arrowheads indicate the loci of mitolysosomes.

**Figure 4 pathogens-13-01139-f004:**
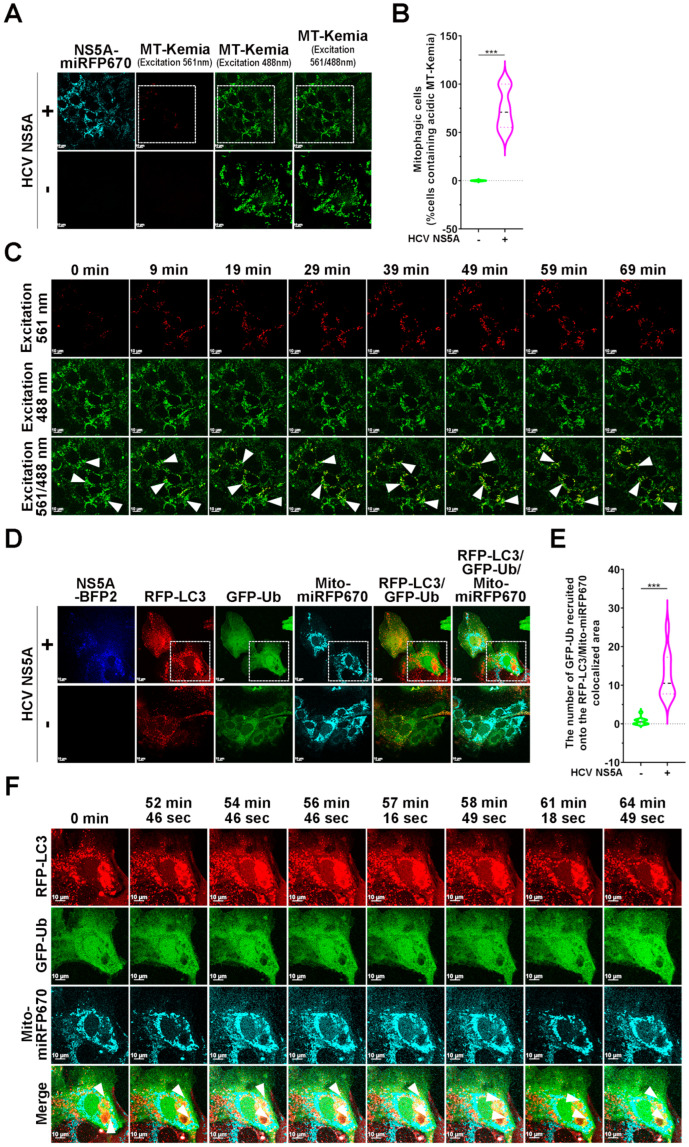
HCV NS5A enhances mitophagic flux and induces ubiquitin recruitment to mitochondria: (**A**) Huh7 cells were transduced with pTRIP-MT-Keima lentiviruses, according to the procedure described in the “Materials and Methods” section, to establish Huh7/MT-Keima cells. Then, Huh7/MT-Keima cells were transduced with (+) or without (−) pTRIP-HCV NS5A-miRFP670 lentiviruses. Forty-eight hours later, the cells were analyzed via confocal microscopy at short (488 nm) and long (561 nm) excitation wavelengths. (**B**) The percentage of cells containing acidic MT-Keima (excitation at 561 nm) was quantified, as described previously [33]. The data are presented as means ± SEMs (*n* = 10, *** *p* < 0.001). (**C**) The selected live imaging frames show the magnified area in the white dashed box of the top panel in (**A**). The white arrowheads indicate the cells expressing acidic MT-Keima. (**D**) Huh7 cells were transduced with lentiviruses expressing RFP-LC3, GFP-Ub, or Mito-miRFP670, as described in the “Materials and Methods” section, generating Huh7/RFP-LC3/GFP-Ub/Mito-miRFP670 cells. Huh7/RFP-LC3/GFP-Ub/Mito-miRFP670 cells were transduced with (+) or without (−) pTRIP-HCV NS5A-mTagBFP2 lentiviruses for forty-eight hours and then analyzed via confocal microscopy. (**E**) The number of RFP-LC3 puncta containing GFP-Ub on Mito-miRFP670-labeled mitochondria was quantified. The data are presented as means ± SEMs (*n* = 10, *** *p* < 0.001). (**F**) The selected live imaging frames show the magnified area in the white dashed box of the top panel in (**A**). The white arrowheads indicate the recruitment of GFP-Ub to Mito-miRFP670-expressing mitochondria and the subsequent sequestration of RFP-LC3-labeled autophagic vacuoles.

**Figure 5 pathogens-13-01139-f005:**
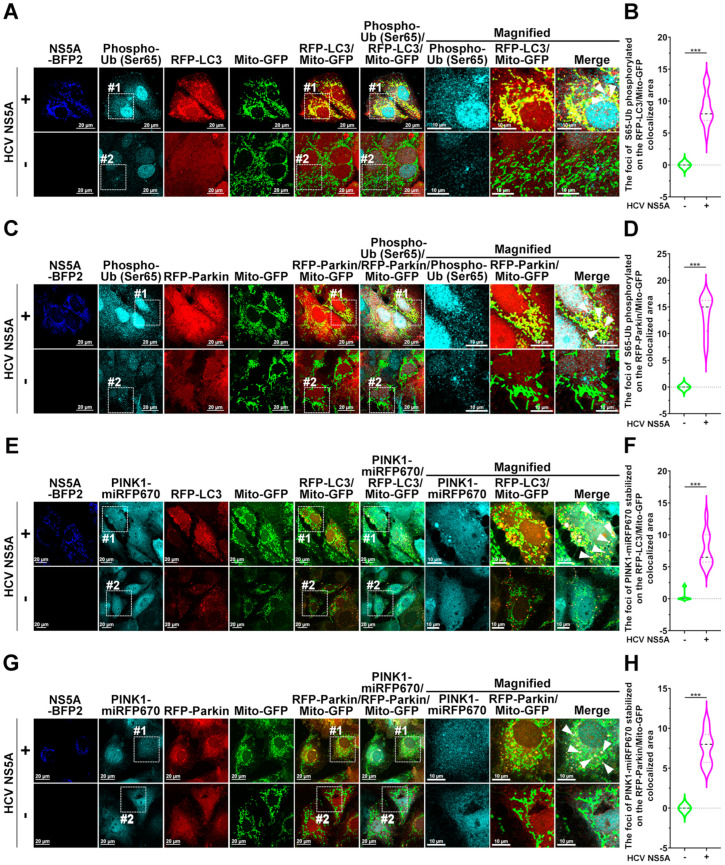
HCV NS5A induces ubiquitin Ser65 phosphorylation and PINK1 stabilization in mitochondria: (**A**,**B**) (**A**) Huh7/RFP-LC3/Mito-GFP cells were transduced with (+) or without (−) pTRIP-HCV NS5A-mTagBFP2 lentiviruses. Forty-eight hours later, the cells were immunostained with a phospho-ubiquitin (Ub; Ser65) antibody and analyzed via confocal microscopy. (**B**) Foci of phospho-ubiquitin (Ser65) recruited onto mitophagosomes in which RFP-LC3 puncta sequestered Mito-GFP-labeled mitochondria were quantified. (**C**,**D**) (**C**) Huh7/RFP-Parkin/Mito-GFP cells were transduced with (+) or without (−) pTRIP-HCV NS5A-mTagBFP2 lentiviruses for forty-eight hours. Then, the cells were immunostained with an anti-phospho-ubiquitin (Ser65) antibody and analyzed via confocal microscopy. (**D**) Foci of phosphor-Ub recruited to RFP-Parkin-translocated Mito-GFP-expressing mitochondria were quantified. (**E**,**F**) (**E**) Huh7/RFP-LC3/Mito-GFP cells were transduced with lentiviruses expressing PINK1-miRFP670, generating Huh7/RFP-LC3/Mito-miRFP670/PINK1-miRFP670 cells. Then, the cells were transduced with (+) or without (−) pTRIP-HCV NS5A-mTagBFP2 lentiviruses for forty-eight hours and analyzed via confocal microscopy. (**F**) Foci of PINK1-miRFP670 stabilized on mitophagosomes, in which RFP-LC3 puncta sequestered Mito-GFP-expressing mitochondria were quantified. (**G**,**H**) (**G**) Huh7/RFP-Parkin/Mito-GFP cells were transduced with lentiviruses expressing PINK1-miRFP670 to establish Huh7/RFP-LC3/Mito-miRFP670/PINK1-miRFP670 cells. Then, the cells were transduced with (+) or without (−) pTRIP-HCV NS5A-mTagBFP2 lentiviruses for forty-eight hours and analyzed via confocal microscopy. (**H**) The number of MiRFP670-PINK1 foci recruited onto the RFP-Parkin-translocated Mito-miRFP670-labeled mitochondria was quantified. The data shown in (**B**,**D**,**F**,**H**) represent the mean ± SEM (*n* = 10, *** *p* < 0.001). The magnified field-1 and magnified field-2 in (**A**,**C**,**E**,**G**) show enlarged images of white dashed boxes 1 and 2 in the top and bottom panels. The white arrowheads indicate colocalized signals.

**Figure 6 pathogens-13-01139-f006:**
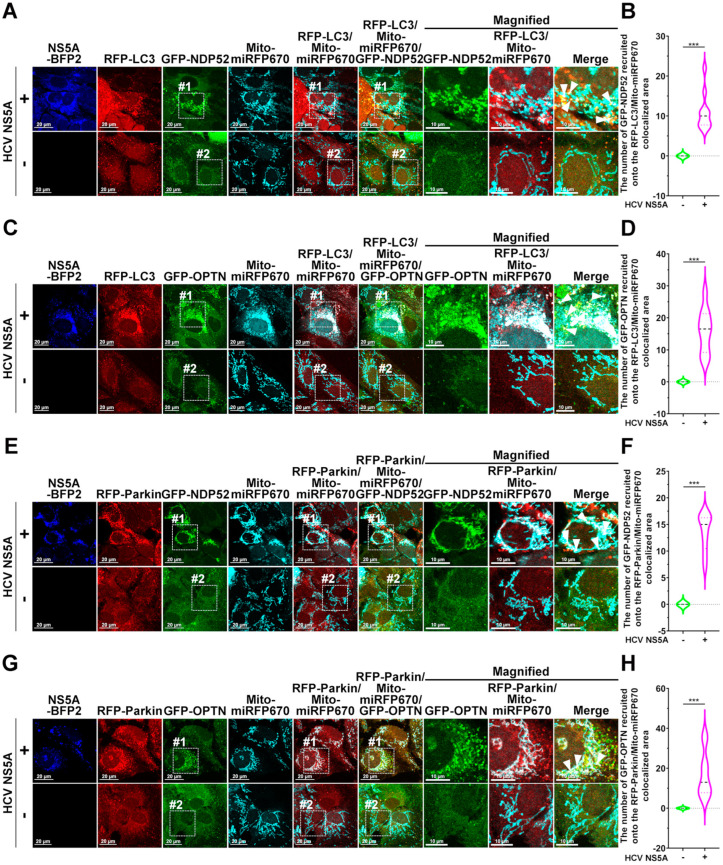
Recruitment of NDP52 and OPTN into HCV NS5A-activated mitophagy: (**A,B**) (**A**) Huh7 cells were transduced with lentiviruses expressing RFP-LC3, Mito-miRFP670, or GFP-NDP52, as described in the “Materials and Methods” section, generating Huh7/RFP-LC3/Mito-miRFP670/GFP-NDP52 cells. Then, the cells were transduced with (+) or without (−) pTRIP-HCV NS5A-mTagBFP2 lentiviruses for forty-eight hours and analyzed via confocal microscopy. (**B**) The number of GFP-NDP52 molecules recruited onto mitophagosomes, in which RFP-LC3 puncta sequestered Mito-miRFP670-expressing mitochondria, was quantified. (**C**,**D**) (**C**) Huh7 cells were transduced with lentiviruses expressing RFP-LC3, Mito-miRFP670, or GFP-OPTN, as described in the “Materials and Methods” section, generating Huh7/RFP-LC3/Mito-miRFP670/GFP-OPTN cells. Then, the cells were transduced with (+) or without (−) pTRIP-HCV NS5A-mTagBFP2 lentiviruses for forty-eight hours and analyzed via confocal microscopy. (**D**) The number of GFP-OPTN molecules recruited onto mitophagosomes, in which RFP-LC3 puncta sequestered Mito-miRFP670-expressing mitochondria, was quantified. (**E**,**F**) (**E**) Huh7/RFP-Parkin/Mito-miRFP670 cells were transduced with lentiviruses expressing GFP-NDP52, generating Huh7/RFP-Parkin/Mito-miRFP670/GFP-NDP52 cells. Then, the cells were transduced with (+) or without (−) pTRIP-HCV NS5A-mTagBFP2 lentiviruses for forty-eight hours and analyzed via confocal microscopy. (**F**) The number of GFP-NDP52 molecules recruited to RFP-Parkin-translocated Mito-miRFP670-labeled mitochondria was quantified. (**G**,**H**) (**G**) Huh7/RFP-Parkin/Mito-miRFP670 cells were transduced with lentiviruses expressing GFP-OPTN to establish Huh7/RFP-Parkin/Mito-miRFP670/GFP-OPTN cells. Then, the cells were transduced with (+) or without (−) pTRIP-HCV NS5A-mTagBFP2 lentiviruses for forty-eight hours and analyzed via confocal microscopy. (H) The number of GFP-OPTN molecules recruited to RFP-Parkin-translocated Mito-miRFP670-labeled mitochondria was quantified. The data shown in (**B**,**D**,**F**,**H**) represent the mean ± SEM (*n* = 10, *** *p* < 0.001). The magnified field-1 and magnified field-2 in (**A**,**C**,**E**,**G**) show enlarged images of white dashed boxes 1 and 2 in the top and bottom panels. The white arrowheads indicate colocalized signals.

**Figure 7 pathogens-13-01139-f007:**
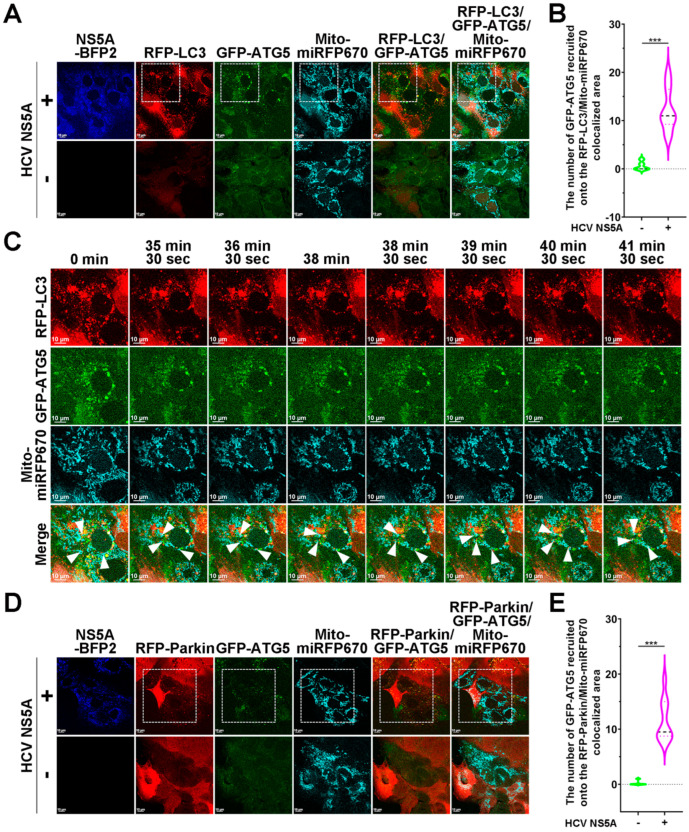
Recruitment of ATG5 into close proximity to mitochondria for HCV NS5A-induced mitophagy: (**A**) Huh7 cells were transduced with lentiviruses expressing RFP-LC3 and mito-miRFP670 to establish Huh7/RFP-LC3/Mito-miRFP670 cells. Huh7/RFP-LC3/Mito-miRFP670 cells were transduced with retroviruses expressing GFP-ATG5, generating Huh7/RFP-LC3/Mito-miRFP670/GFP-ATG5 cells. Then, Huh7/RFP-LC3/Mito-miRFP670/GFP-ATG5 cells were transduced with (+) or without (−) pTRIP-HCV NS5A-mTagBFP2 lentiviruses for forty-eight hours and analyzed via confocal microscopy. (**B**) The number of GFP-ATG5 molecules recruited onto mitophagosomes, in which RFP-LC3 puncta sequestered Mito-miRFP670-expressing mitochondria, was quantified. (**C**) The selected live imaging frames show the magnified area in the white dashed box of the top panel in (**A**). The white arrowheads indicate the recruitment of GFP-ATG5 to Mito-miRFP670-labeled mitochondria before sequestration by RFP-LC3 puncta. (**D**) Huh7 cells were transduced with lentiviruses expressing RFP-Parkin and Mito-miRFP670 to establish Huh7/RFP-Parkin/Mito-miRFP670 cells. Huh7/RFP-Parkin/Mito-miRFP670 cells were transduced with retroviruses expressing GFP-ATG5, generating Huh7/RFP-Parkin/Mito-miRFP670/GFP-ATG5 cells. Then, Huh7/RFP-Parkin/Mito-miRFP670/GFP-ATG5 cells were transduced with (+) or without (−) pTRIP-HCV NS5A-mTagBFP2 lentiviruses for forty-eight hours and analyzed via confocal microscopy. (**E**) The number of GFP-ATG5 molecules recruited to RFP-Parkin-translocated Mito-miRFP670-labeled mitochondria was quantified. (**F**) The selected live imaging frames show the magnified area in the white dashed box of the top panel in (**D**). The white arrowheads indicate the recruitment of GFP-ATG5 to Mito-miRFP670-labeled mitochondria after translocation by RFP-Parkin. The data shown in (**B**,**E**) represent the mean ± SEM (*n* = 10, *** *p* < 0.001).

**Figure 8 pathogens-13-01139-f008:**
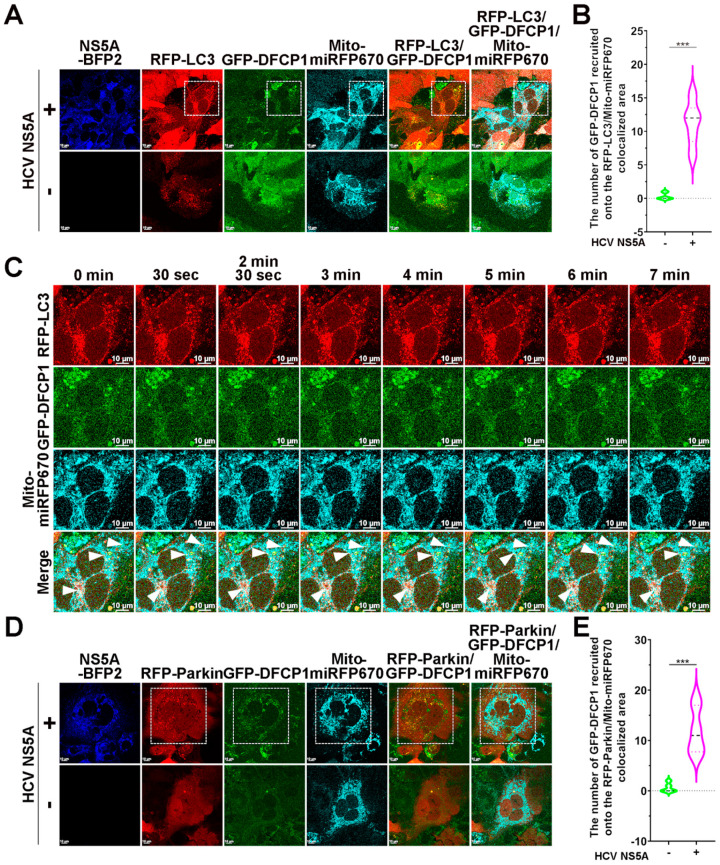
Recruitment of DFCP1 into the proximity of mitochondria for HCV NS5A-induced mitophagy: (**A**) Huh7/RFP-LC3/Mito-miRFP670 cells were transduced with retroviruses expressing GFP-DFCP1, generating Huh7/RFP-LC3/Mito-miRFP670/GFP-DFCP1 cells. Then, Huh7/RFP-LC3/Mito-miRFP670/GFP-DFCP1 cells were transduced with (+) or without (−) pTRIP-HCV NS5A-mTagBFP2 lentiviruses for forty-eight hours and analyzed via confocal microscopy. (**B**) The number of GFP-DFCP1 molecules recruited onto mitophagosomes, in which RFP-LC3 puncta sequestered Mito-miRFP670-expressing mitochondria, was quantified. (**C**) The frames of selected live images show the magnified area in the white dashed box of the top panel in (**A**). The white arrowheads indicate the recruitment of GFP-DFCP1 to Mito-miRFP670-labeled mitochondria before sequestration by RFP-LC3 puncta. (**D**) Huh7/RFP-Parkin/Mito-miRFP670 cells were transduced with retroviruses expressing GFP-DFCP1, generating Huh7/RFP-Parkin/Mito-miRFP670/GFP-DFCP1 cells. Then, Huh7/RFP-Parkin/Mito-miRFP670/GFP-DFCP1 cells were transduced with (+) or without (−) pTRIP-HCV NS5A-mTagBFP2 lentiviruses for forty-eight hours and analyzed via confocal microscopy. (**E**) The number of GFP-DFCP1 molecules recruited to RFP-Parkin-translocated Mito-miRFP670-labeled mitochondria was quantified. (**F**) The frames of selected live images show the magnified area in the white dashed box of the top panel in (**D**). The white arrowheads indicate the recruitment of GFP-DFCP1 to Mito-miRFP670-labeled mitochondria after translocation by RFP-Parkin. The data shown in (**B**,**E**) represent the mean ± SEM (*n* = 10, *** *p* < 0.001).

**Figure 9 pathogens-13-01139-f009:**
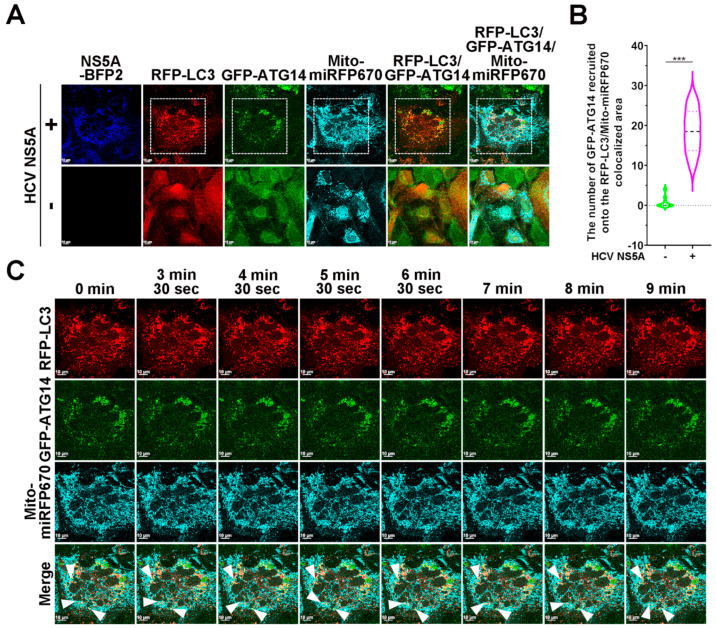
Translocation of ATG14 into close proximity to mitochondria for HCV NS5A-induced mitophagy: (**A**) Huh7/RFP-LC3/Mito-miRFP670 cells were transduced with retroviruses expressing GFP-ATG14, generating Huh7/RFP-LC3/Mito-miRFP670/GFP-ATG14 cells. Then, Huh7/RFP-LC3/Mito-miRFP670/GFP-ATG14 cells were transduced with (+) or without (−) pTRIP-HCV NS5A-mTagBFP2 lentiviruses for forty-eight hours and analyzed via confocal microscopy. (**B**) The number of GFP-ATG14 molecules recruited onto mitophagosomes, in which RFP-LC3 puncta sequester Mito-miRFP670-expressing mitochondria, was quantified. (**C**) The selected live imaging frames show the magnified area in the white dashed box of the top panel in (**A**). The white arrowheads indicate the recruitment of GFP-ATG14 to Mito-miRFP670-labeled mitochondria before sequestration by RFP-LC3 puncta. (**D**) Huh7/RFP-Parkin/Mito-miRFP670 cells were transduced with retroviruses expressing GFP-ATG14, generating Huh7/RFP-Parkin/Mito-miRFP670/GFP-ATG14 cells. Then, Huh7/RFP-Parkin/Mito-miRFP670/GFP-ATG14 cells were transduced with (+) or without (−) pTRIP-HCV NS5A-mTagBFP2 lentiviruses for forty-eight hours and analyzed via confocal microscopy. (**E**) The number of GFP-ATG14 molecules recruited to RFP-Parkin-translocated Mito-miRFP670-labeled mitochondria was quantified. (**F**) The selected live imaging frames show the magnified area in the white dashed box of the top panel in (**D**). The white arrowheads indicate the recruitment of GFP-ATG14 to Mito-miRFP670-labeled mitochondria after RFP-Parkin translocation. The data shown in (**B**,**E**) represent the mean ± SEM (*n* = 10, *** *p* < 0.001).

**Figure 10 pathogens-13-01139-f010:**
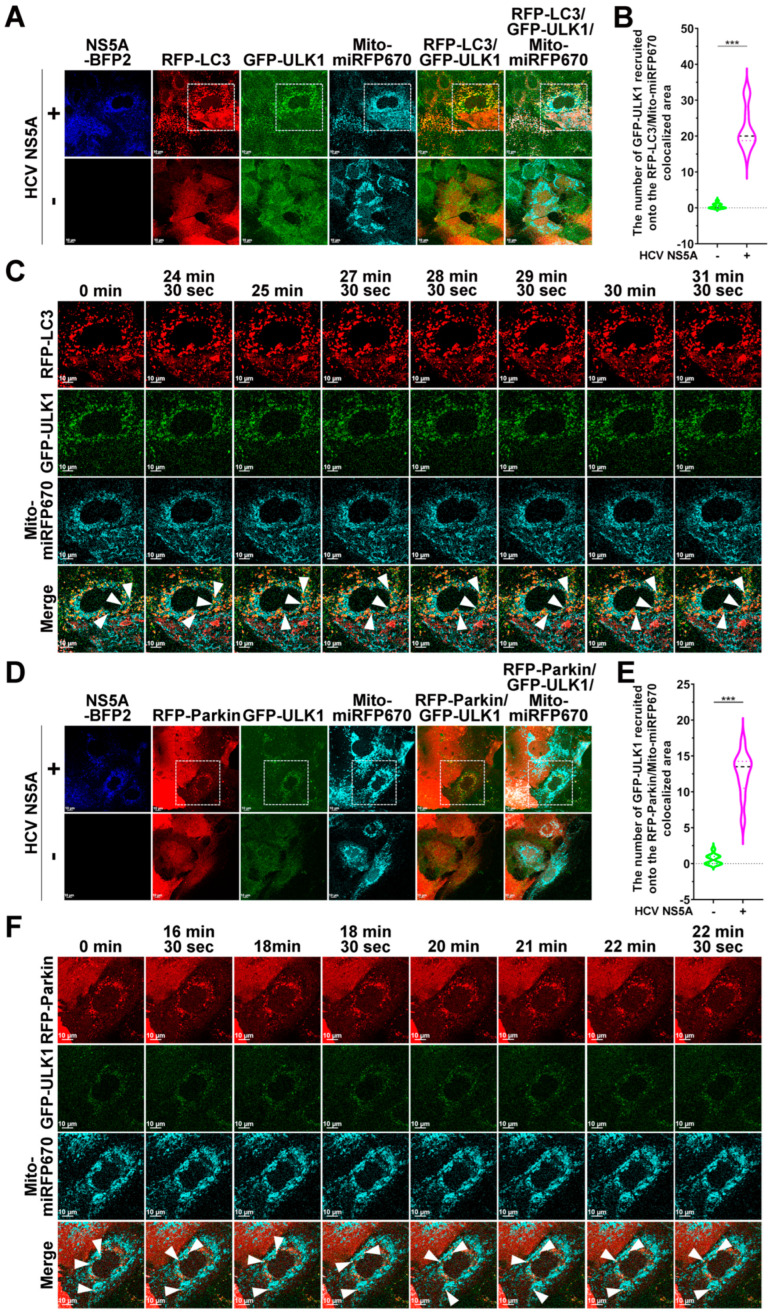
Translocation of ULK1 into the proximity of mitochondria for HCV NS5A-activated mitophagy: (**A**) Huh7/RFP-LC3/Mito-miRFP670 cells were transduced with retroviruses expressing GFP-ULK1, generating Huh7/RFP-LC3/Mito-miRFP670/GFP-ULK1 cells. Then, Huh7/RFP-LC3/Mito-miRFP670/GFP-ULK1 cells were transduced with (+) or without (−) pTRIP-HCV NS5A-mTagBFP2 lentiviruses for forty-eight hours and analyzed via confocal microscopy. (**B**) The number of GFP-ULK1 molecules recruited onto mitophagosomes, in which RFP-LC3 puncta sequestered Mito-miRFP670-expressing mitochondria, was quantified. (**C**) The selected live imaging frames show the magnified area in the white dashed box of the top panel in (**A**). The white arrowheads indicate the recruitment of GFP-ULK1 to Mito-miRFP670-labeled mitochondria before sequestration by RFP-LC3 puncta. (**D**) Huh7/RFP-Parkin/Mito-miRFP670 cells were transduced with retroviruses expressing GFP-ULK1, generating Huh7/RFP-Parkin/Mito-miRFP670/GFP-ULK1 cells. Then, Huh7/RFP-Parkin/Mito-miRFP670/GFP-ULK1 cells were transduced with (+) or without (−) pTRIP-HCV NS5A-mTagBFP2 lentiviruses for forty-eight hours and analyzed via confocal microscopy. (**E**) The number of GFP-ULK1 molecules recruited to the RFP-Parkin-translocated Mito-miRFP670-labeled mitochondria was quantified. (**F**) The selected live imaging frames show the magnified area in the white dashed box of the top panel in (**D**). The white arrowheads indicate the recruitment of GFP-ULK1 to Mito-miRFP670-labeled mitochondria after RFP-Parkin translocation. The data shown in (**B**,**E**) represent the mean ± SEM (*n* = 10, *** *p* < 0.001).

## Data Availability

Data available on request from the authors.

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
