# Peer review of "Hepatitis C Virus NS5A Activates Mitophagy Through Cargo Receptor and Phagophore Formation"

_pathogens, 2024, doi:10.3390/pathogens13121139_

Round 1

Reviewer 1 Report

Comments and Suggestions for Authors

Hsiao CY and coworkers from Taiwan conducted a nice piece of basic research. 

These authors investigated the activity of HCV NS5A, a multi-functional 

protein that regulates the life cycle of HCV.

Various steps have been addressed by Hsiao and coworkers:

1) HCV NS5A promotes the sequestration of mitochondria within autophagic vacuoles; 2) HCV NS5A supports the recruitment of PINK1/Parkin to mitochondria; 3) HCV NS5A helps mitophagy with the aim to make degradation of mitochondria; 4) HCV NS5A stimulates the recruitment of ubiquitin to mitochondria; 5) HCV NS5A supports mitophagy associated with PINK1/Parkin pathway; 6) In HCV NS5A-associated mitophagy the production of autophagosomes occurs (near to michondria); 7) In HCV NS5A-associated mitophagy there is the maturation of phagophores into autophagosomes 

The paper is well written,

The number of references is clearly excessive (n=192). It is not a narrative or systematic review

The section Intoduction (n=3 pages) is too long, the meaning of the terms PINK1 and Parkin remains unclear, the number of references (n=192) is excessive

The evidence shown in page 3 (lines 123-145) is redundant and I suggest to delete it

Author Response

Dear Reviewer:

Thank you for giving us the opportunity to resubmit my manuscript entitled ‶Hepatitis C Virus NS5A Activates Mitophagy through Cargo Receptor and Phagophore Formation″ to Pathogens, (Manuscript ID: pathogens-3349100). We appreciate the thoughtful and constructive suggestions provided by the reviewer. The content of this manuscript has been strengthened according to the reviewer's comments. The revised manuscript shows the changes and point-by-point responses to each comment, which are listed below.

Point 1: The number of references is clearly excessive (n=192). It is not a narrative or systematic review.

Response 1: We thank the reviewer for the thoughtful comments on our manuscript. We have reorganized the cited references and vastly reduced the number of references in the revised manuscript. Please see line 898 on page 29 to line 1086 on page 33 in the revised manuscript. 

Point 2: The section Intoduction (n=3 pages) is too long, the meaning of the terms PINK1 and Parkin remains unclear, the number of references (n=192) is excessive.

Response 2: We are very grateful for the reviewer's thoughtful comment. In the revised manuscript, we have shortened the content of the ‶Introduction″ section. Please see line 27 on page 1 to line 121 on page 3 in the revised manuscript. We have clarified the terms of PINK1 and Parkin in the revised manuscript. Please see lines 57~60 on page 2 in the revised manuscript. The number of references in the revised manuscript has mainly been reduced. Please see line 898 on page 29 to line 1086 on page 33 in the revised manuscript.

Point 3: The evidence shown in page 3 (lines 123-145) is redundant and I suggest to delete it.

Response 3: We appreciate the reviewer's suggestions. This part has been removed from the revised manuscript. It is hoped that the review kindly agrees with us to keep a summary of conclusions for our study at the end of the ‶Introduction″ section. Please see lines 117~121 on page 3 in the revised manuscript.

We hope that this version of our manuscript and our responses address all your concerns and that this revised manuscript meets the criteria for publication in Pathogens. Thank you for your kind consideration.

Sincerely,

Po-Yuan Ke, Ph.D.

Associate Professor

Department of Biochemistry & Molecular Biology and Graduate Institute of Biomedical Sciences, College of Medicine, Chang Gung University, Taoyuan 33302, Taiwan, Republic of China

Liver Research Center, Chang Gung Memorial Hospital, Linkou, Taoyuan 33305, Taiwan, Republic of China

Tel: 886-3-2118800-5115

E-mail: pyke0324@mail.cgu.edu.tw

Reviewer 2 Report

Comments and Suggestions for Authors

This is a detailed molecular paper on Hepatitis C Virus NS5A activating mitophagy through cargo receptor recognition and proximal Formation of Phagophore. The authors conclude that HCV NS5A induces PINK1/Parkin-dependent mitophagy reconizig mitochondria by cargo receptors and the nascent formation of phagophores close to mitochondria.
Minor comment:

1.      A graphical abstract would be great to have rapid view or the technical aspects and the molecular events leading to phagophore formation.

2.      Abbreviations: Please add a list of abbreviations.

3.      Title: Reconsider a shorter version.

Comments on the Quality of English Language

OK

Author Response

Dear Reviewer:

Thank you for giving us the opportunity to resubmit my manuscript entitled ‶Hepatitis C Virus NS5A Activates Mitophagy through Cargo Receptor and Phagophore Formation″ to Pathogens, (Manuscript ID: pathogens-3349100). We appreciate the thoughtful and constructive suggestions provided by the reviewer. The content of this manuscript has been strengthened according to the reviewer's comments. The revised manuscript shows the changes and point-by-point responses to each comment, which are listed below.

Point 1: A graphical abstract would be great to have rapid view or the technical aspects and the molecular events leading to phagophore formation.

Response 1: We thank the reviewer for the thoughtful suggestions on our manuscript. We have incorporated a graphical abstract into the resubmission of our manuscript. Please see the graphical abstract in our resubmission files.

Point 2: Abbreviations: Please add a list of abbreviations.

Response 2: We are grateful for the reviewer's comment. We have added a list of abbreviations in the revised manuscript. Please see lines 840~867 on page 28 in the revised manuscript.

Point 3: Title: Reconsider a shorter version.

Response 3: Thank you very much for this suggestion. We have revised the title of our revised manuscript to a shorter version, entitled ‶Hepatitis C Virus NS5A Activates Mitophagy through Cargo Receptor and Phagophore Formation″. Please see Page 1 in the revised manuscript.

We hope that this version of our manuscript and our responses address all your concerns and that this revised manuscript meets the criteria for publication in Pathogens. Thank you for your kind consideration.

Sincerely,

Po-Yuan Ke, Ph.D.

Associate Professor

Department of Biochemistry & Molecular Biology and Graduate Institute of Biomedical Sciences, College of Medicine, Chang Gung University, Taoyuan 33302, Taiwan, Republic of China

Liver Research Center, Chang Gung Memorial Hospital, Linkou, Taoyuan 33305, Taiwan, Republic of China

Tel: 886-3-2118800-5115

E-mail: pyke0324@mail.cgu.edu.tw

Reviewer 3 Report

Comments and Suggestions for Authors

The manuscript studies how the HCV NS5A protein triggers mitophagy through the PINK1/Parkin pathway. The authors use advanced imaging and molecular methods to provide insights into how the virus affects cells. While the paper is strong and worth publishing, a few points should be addressed:

 1.   The paper mentions that HCV-induced mitochondrial degradation could disrupt MAVS signaling (Line 832). It would be helpful to explore this further in the current study.
  2.  Full-length HCV produces other proteins. Could these proteins counteract NS5A's role? Showing that other HCV proteins do not trigger the same response would make the findings stronger.
  3.  Parts of Figures 5 and 6 are too small to see clearly. Enlarging these sections would make them easier to understand.
  4. Adding NS5A mutants that cannot cause mitophagy or using PINK1/Parkin inhibitors as controls would improve the experiments and make the results more reliable.

Addressing these points would make the manuscript even better.

Author Response

Dear Reviewer:

Thank you for giving us the opportunity to resubmit my manuscript entitled ‶Hepatitis C Virus NS5A Activates Mitophagy through Cargo Receptor and Phagophore Formation″ to Pathogens, (Manuscript ID: pathogens-3349100). We appreciate the thoughtful and constructive suggestions provided by the reviewer. The content of this manuscript has been strengthened according to the reviewer's comments. The revised manuscript shows the changes and point-by-point responses to each comment, which are listed below.

Point 1: The paper mentions that HCV-induced mitochondrial degradation could disrupt MAVS signaling (Line 832). It would be helpful to explore this further in the current study.

Response 1: We thank the reviewer for the thoughtful comments on our manuscript. In the ‶Discussion″ section of our manuscript, we discussed that influenza A virus PB1-F2 and nucleoprotein have been shown to induce mitophagy to promote MAVS degradation, thus suppressing type I interferon (IFN) response (references 80-81 in the revised manuscript) (please see lines 799~802 in the revised manuscript). However, it remains unknown whether HCV NS5A activates mitophagy to repress innate anti-HCV immunity by promoting the degradation of MAVS. Accordingly, we will plan to investigate if HCV NS5A leads to MAVS degradation via mitophagy to inhibit anti-HCV IFN response in the future study (please see lines 802~806 in the revised manuscript). Our current study shows that ectopic expression of HCV NS5A induces the downregulation of MAVS (please see the attached Figure 1). In the next study, we will study whether mitophagy functions in the HCV NS5A-induced MAVS degradation in mitophagy-competent and deficient cells. Moreover, we will investigate if the HCV NS5A-induced MAVS degradation represses type I IFN response. It is hoped that the reviewer kindly agrees with us to keep this comment for future direction in our next study. Our current manuscript aims to dissect the molecular mechanisms underlying HCV NS5A-activated mitophagy, including cargo receptor recognition and proximal formation of phagophore. Again, we appreciate your kind comments on the future direction of our study.

Figure 1. Induced MAVS downregulation by HCV NS5A: Huh7 cells were transduced with (+) and without (-) pTRIP-HCV NS5A-3XFLAG. Forty-eight hours later, the cells were harvested and analyzed for protein expressions via SDS-PAGE and western blotting.

Point 2:  Full-length HCV produces other proteins. Could these proteins counteract NS5A's role? Showing that other HCV proteins do not trigger the same response would make the findings stronger.

Response 2: We are grateful for the reviewer's suggestion. We have incorporated additional experiments to demonstrate that HCV core and HCV NS5B cannot promote mitophagic degradation in MT-Keima reporter cells, compared to the induced mitophagic degradation by HCV NS5A in MT-Keima reporter cells (Supplemental Figures 3A and 3B in the revised manuscript). These results demonstrate that HCV NS5A specifically activates mitophagy to promote mitochondrial degradation. Please see Supplemental Figures 3A and 3B and lines 418~419 on page 13 in the revised manuscript.

Point 3: Parts of Figures 5 and 6 are too small to see clearly. Enlarging these sections would make them easier to understand.

Response 3: Thank you very much for this comment. We have revised Figures 5 and 6, particularly for the enlargement of magnified figures in the revised manuscript. Please see Figure 5 on pages 15~16 and Figure 6 on pages 17~18 in the revised manuscript.

Point 4: Adding NS5A mutants that cannot cause mitophagy or using PINK1/Parkin inhibitors as controls would improve the experiments and make the results more reliable.

Response 4: We appreciate the reviewer's suggestion. The mitophagy inhibitor, specifically for the inhibitor of PINK1 and Parkin, is not currently available. The mitochondrial division inhibitor-1 (Mdivi-1) is the only inhibitor that has been shown to suppress mitophagy (reference 47 in the revised manuscript). Therefore, we studied whether Mdivi-1 treatment can inhibit HCV NS5A-induced mitophagy. In Supplemental Figures 3C and 3D of the revised manuscript, we found that Mdivi-1 dramatically repressed the HCV NS5A-enhanced mitophagic degradation in MT-Keima reporter cells. In addition, gene knockdown of endogenous Parkin by RNA interference inhibited HCV NS5A-activated increase in mitophagic degradation in MT-Keima reporter cells (Supplemental Figures 3E and 3F in the revised manuscript). These results imply that HCV NS5A specifically activates Parkin-dependent mitophagy to promote mitochondrial turnover. Please see Supplemental Figures 3C-D and 3E-F and lines 419~421 on page 13 in the revised manuscript. Moreover, gene knockout of NDP52 and OPTN, two mitophagy receptors, significantly suppressed the HCV NS5A-induced mitophagic degradation (Supplemental Figure 6 in the revised manuscript). In our next study, we will dissect which domain of HCV NS5A functions in mitophagy activation and mitochondrial turnover. We discussed this future direction in the ‶Discussion″ section of our manuscript (please see line 777 on page 26 to line 787 on page 27 in the revised manuscript).

We hope that this version of our manuscript and our responses address all your concerns and that this revised manuscript meets the criteria for publication in Pathogens. Thank you for your kind consideration.

Sincerely,

Po-Yuan Ke, Ph.D.

Associate Professor

Department of Biochemistry & Molecular Biology and Graduate Institute of Biomedical Sciences, College of Medicine, Chang Gung University, Taoyuan 33302, Taiwan, Republic of China

Liver Research Center, Chang Gung Memorial Hospital, Linkou, Taoyuan 33305, Taiwan, Republic of China

Tel: 886-3-2118800-5115

E-mail:
